# Development and characterisation of novel oxytocin analogues for PET imaging
Giancarlo Pascali [1,2,3] ✉, Arvind Parmar [1,3] ✉, Simone Zanoni[2], Andrew Arthur[1], Anke Hering[4], Ngari Teakle[4], Jack Markham[1,3], Bo Zhang[1,5], Tiffany Mackay[3], Mitch Klenner[1,6], Lawson Spare [1,7], Ivan Greguric[1], Amanda McDonald[1], Aleksandra Bjelosevic [1], Lidia Matesic[1], Gita Rahardjo[1], David Zahra[1], Hasar Hamze[1], Ian B. Hickie [3], Richard B. Banati[1,3], Marie-Claude Gregoire[1,3,8], Larry Young [9], Markus Muttenthaler [4,10] ✉ & Adam J. Guastella [3,11] ✉

The oxytocin/oxytocin receptor (OT/OTR) signalling system is involved in socioemotional behaviours, garnering interest as a therapeutic target across multiple clinical conditions. Despite its potential, our limited understanding of how to optimally target it and the scarcity of molecular tools for in vivo studies hinder therapeutic development. Molecular imaging techniques, such as Positron Emission Tomography (PET), can bridge this gap by furnishing direct insights into ligand biodistribution, receptor visualisation and ligand-receptor engagement. Here, we report the design, synthesis and biochemical and pharmacological characterisation of five OT-like peptides as novel PET tracers for investigating the OT/OTR signalling system. dOTK[8][SFB] emerged as the most promising OT-like lead. The radioactive version [[18]F]dOTK[8][SFB] was produced using a microfluidic reaction approach and validated by preclinical PET imaging of healthy rats after intravenous ligand administration. [[18]F]dOTK[8][SFB] exhibited specific accumulation in OTR-rich tissues, affirming OTR-specificity and suitability as a new OT-like PET radiotracer for investigating OT/OTR biodistribution in humans.

The oxytocin/oxytocin receptor (OT/OTR) signalling system is intricately involved in numerous physiological functions in humans, encompassing reproductive processes such as labour induction and progression, lactation, and ejaculation, as well as a spectrum of central behaviours including empathy, trust, satiety, memory, learning, motivation, reward, anxiety, and stress response[1,2]. OT has found extensive clinical use in promoting uterine contractions during labour and facilitating milk let-down during breastfeeding. Over the past three decades, the research focus has shifted towards understanding OT's role in social behaviours across mammalian species, including bonding, social cognition, and social development[3–5]. OT is critical in fostering pair-bond formation, nurturing behaviours, social recognition, and memory, which are crucial components in early social development and the acquisition of social skills[6–9]. Notably, OT administration to children and adults *via* intranasal or intravenous routes has shown promise in enhancing various social

behaviours and functions[10,11], having spurred investigations into OT as a therapeutic agent for such conditions.

Intravenous OT administration exhibits reliable effects in supporting uterine contractions during labour, yet the intranasal administration of OT for other therapeutic applications remains highly debated due to reproducibility and effectiveness concerns[12]. Nonetheless, intranasal administration is the preferred route for experimental and therapeutic purposes outside the hospital, due to its ease of use and purported and highly debated greater brain penetration[13]. OT brain penetration has been investigated in several animal studies[9,14], including non-human primates[15], but investigations in humans have been limited to indirect methods such as quantifying blood flow post-administration, thus not providing a clear consensus[16]. Molecular imaging techniques, such as Positron Emission Tomography (PET), offer direct avenues for analysis extending to human subjects, and would be highly relevant for studying OT route of administration, biodistribution,

[1]Australian Nuclear Science and Technology Organisation, Sydney, NSW, Australia. [2]School of Chemistry, University of New South Wales, Sydney, NSW, Australia. [3]Brain and Mind Centre, Faculty of Medicine and Health, University of Sydney, Sydney, NSW, Australia. [4]Institute for Molecular Bioscience, The University of Queensland, Brisbane, QLD, Australia. [5]Rad/Molecular Imaging Program, Stanford University, Stanford, CA, USA. [6]Quantum Pharma Australia, Doncaster, VIC, Australia. [7]Department of Nuclear Medicine, Liverpool Hospital, Liverpool, NSW, Australia. [8]Canadian Nuclear Laboratories, Chalk River, ON, Canada. [9]Emory University, Atlanta, GA, USA. [10]Institute of Biological Chemistry, Faculty of Chemistry, University of Vienna, Vienna, Austria. [11]Children's Hospital Westmead Clinical School, Faculty of Medicine and Health, University of Sydney, Sydney, NSW, Australia. ✉e-mail: gianp@ansto.gov.au; arvind.parmar@sydney.edu.au; markus.muttenthaler@univie.ac.at; adam.guastella@sydney.edu.au

brain uptake and visualisation of OT-OTR engagement. However, the lack of suitable OT radiotracers for human PET imaging has hindered progress in this area[17].

Early attempts to develop radiolabelled OT analogues, such as [K([$^{111}$In]In-DOTA)]$^8$dOT[18] and [Gly-(2-[$^{18}$F]fluoroethyl)NH$^9$]OT[19] showed promise but faced challenges in radiochemistry scale and in vivo imaging, and lacked comprehensive pharmacokinetic and pharmacodynamic characterisation. A more recent attempt with [K([$^{18}$F]AlF-NODA)]$^8$dOT, demonstrated favourable affinity and selectivity for OTR but fell short in brain parenchyma penetration as the desired target[20]. Non-peptidic, small-molecule OTR PET tracers have also been explored, albeit with limited success[21–28].

Therefore, the aim of this study was to develop a suitable PET radiotracer for assessing and tracking OT administration in vivo. We wanted to develop an analogue closely mimicking the physicochemical and pharmacological properties of endogenous OT, which would allow us to use this OT-like PET tracer to answer fundamental research questions regarding its biodistribution in humans and help identify any differences between intranasal and intravenous routes of administration. Tracer development included the design, synthesis and physiochemical and pharmacological characterisation of OT-like analogues amenable to radiofluorination, and comparison and optimisation of various fluorine-18 labelling approaches. We finally validated the suitability of our lead tracer for PET imaging using biodistribution, dynamic imaging and OTR specificity data, setting the scene for future human studies.

## Results and discussion
### Design, synthesis and biochemical characterisation of different OT analogues

We designed, synthesised, and biochemically characterised five **OT** analogues with the aim of identifying an OT-like candidate for fluorine-18 labelling and biodistribution studies. We selected structures suitable for radiolabelling using well-established bioconjugation strategies and compared their biochemical and pharmacological properties with endogenous OT.

**Chemical synthesis of OT analogues.** The first two OT analogues were derived from N-terminal desamino-OT (dOT) and included a Leu$^8$-to-Lys$^8$ modification (dCys$^1$,Lys$^8$-OT or **dOTK$^8$**) to introduce a handle for the fluorine-18. Lys$^8$ is a common modification well tolerated regarding OTR affinity, even when functionalized with different reporter tags[29]. Removal of the N-terminal amino group (dCys$^1$) is also well tolerated and facilitates Lys$^8$ labelling, as the sidechain of Lys$^8$ becomes the only free amino group available. Using dOTK$^8$ as the parent peptide, we then introduced two different fluorinated prosthetic groups to Lys$^8$ through amide bond formation or Michael addition (Fig. 1A). In addition, we pursued the bioorthogonal oxime ligation through instalment of an amino-oxy group at Lys$^8$ (**OTK$^8$[Aoa]**) to enable the incorporation of two additional fluorinated motifs to OT. The biorthogonality of the oxime ligation allowed for keeping the N-terminal amino group of OT, which was of interest to us as we wanted to closely mimic OT. These approaches resulted in five different OT-like analogues (Fig. 1A).

**dOTK$^8$** was synthesised through Fmoc-SPPS (fluorenylmethoxycarbonyl-mediated solid-phase peptide synthesis), followed by global peptide deprotection and resin cleavage using TFA (trifluoracetic acid) and subsequent oxidative folding to form the single disulphide bond between dCys$^1$ and Cys$^6$. **OTK$^8$[Aoa]** was synthesised via a similar SPPS procedure, as described in the experimental section. The fluorobenzoate group was introduced by reacting N-succinimidyl-4-fluorobenzoate (**SFB**) with the ε-amino group of Lys$^8$, forming an amide bond and yielding **dOTK$^8$[SFB]** in quantitative yield. The ESF-containing analogues **dOTK$^8$[ESF]** and **dOTK$^8$[ESF$_2$]** were produced via a Michael addition, resulting in mono- or di-substituted analogues through equimolar or excess amount of reagent, respectively. Functionalization yields were only 10% and 30% for the mono- and di-substituted analogues, respectively. **OTK$^8$[Aoa-FBA]** and

**OTK$^8$[Aoa-FDG]** were produced via oxime ligation of 4-fluorobenaldehyde (**FBA**) or 2-deoxyfluoroglucose (**FDG**), respectively. Since the amino-oxy group was unstable during oxidative folding, we pre-formed the oxime ligation before disulphide bond formation. The yields obtained from this 2-step functionalization process were high, with 100% for **OTK$^8$[Aoa-FBA]** and 85% for **OTK$^8$[Aoa-FDG]**. While this approach produced the desired standards for characterisation purposes, another labelling strategy would be needed for radiofluorination due to its short half-life (109.7 min). A faster strategy has recently been developed using non-radioactive fluorinated compounds but has not been applied to fluorine-18[30].

**Hydrophobicity assessment of synthesised analogues.** To identify the most OT-like radiotracer, we first assessed the hydrophobicity of all compounds by comparing their retention times with **OT** on an analytical C$_{18}$-RP-HPLC column (Fig. 2A). This revealed the following hydrophobicity distribution, starting with the least hydrophobic one: OTK$^8$[Aoa-FDG] < dOTK$^8$[ESF] < OT < dOTK$^8$[SFB] < dOTK$^8$[ESF$_2$] < OTK$^8$[Aoa-FBA]; this order is slightly different from the one obtained using cLogP values, as it takes into account secondary structure, hydrogen bonding, disulphide macrocyclization, charge distribution and buried residues. While OTK$^8$[Aoa-FDG], dOTK$^8$[ESF], and dOTK$^8$[SFB] had similar retentions times as OT, dOTK$^8$[ESF$_2$] and OTK$^8$[Aoa-FBA] were significantly more hydrophobic, which could markedly affect biodistribution compared to OT.

**Stability assessment of synthesised analogues.** First, candidate analogues were exposed to buffered solutions at pH 3, pH 7, and pH 9 to assess their pH stability, using analytical C$_{18}$-RP-HPLC over a period of 10 h. At these pH conditions, all the analogues were stable with half-lives ($t_{1/2}$) > 10 h, except for **dOTK$^8$[ESF$_2$]** ($t_{1/2}$ = 3.6 h at pH 3, $t_{1/2}$ < 30 min at pH 7 and 9). Some degradation occurred within 10 h for **dOTK$^8$[SFB]** and **OTK$^8$[Aoa-FDG]** at pH 7 (25%, *SI-1*). These results, along with the low yields of ESF functionalization, led to the decision to not pursue the Michael addition route for functionalization.

OT is metabolised in vivo by oxytocinases, a subfamily of M1 peptidases consisting of P-LAP/IRAP, ERAP1, and ERAP2 in the blood[31]. Therefore, we assessed the metabolic half-life of our candidates in rat and human serum over 48 h using analytical RP-HPLC (Figs. 2B and SI-2). All tested analogues exhibited half-lives greater than 12 hours, sufficient for PET imaging purposes for both rat and human studies, where tracer distribution is typically studied for up to 2 h. The main metabolite identified by LC-MS analysis was the dCys/Cys-Tyr-Ile-Gln-Asn-Cys-Pro macrocycle (*SI-3*), based on matching mass and reasonable C-terminal enzymatic cleavage.

Even if these stability assessments have been performed over several hours on model systems, and such durations are not applicable to fluorine-18, we believe that such experimental strategy would provide radiopharmaceuticals that are more stable at the tracer levels used in PET and real-life in vivo scenarios.

**Assessment of targeting and selectivity of selected synthesised analogues.** OTR belongs to the rhodopsin-type (class I) G protein-coupled receptor (GPCR) family and is closely related to the vasopressin (VP) receptor (VPR) family (V$_{1a}$R, V$_{1b}$R, V$_2$R). While OT is the native ligand for OTR, it also displays cross-reactivity at the VPRs[32]. OTR, V$_{1a}$R, and V$_{1b}$R are expressed in the periphery and in CNS, while V$_2$R is primarily expressed in the kidneys and was deemed not in scope for our exploration[33]. The EC$_{50}$ of our ligands was measured using a Fluorescent Imaging Plate Reader (FLIPR) calcium mobilisation assay (Fig. 2C)[34].

All tested candidates retained a low nanomolar potency at OTR (5–47 nM), a range consistent with values reported for other validated PET tracers. **OTK$^8$[Aoa-FDG]** was slightly more potent (~2-fold) than OT at OTR, while **dOTK$^8$[SFB]** and **OTK$^8$[Aoa-FBA]** were less potent (~4-fold).

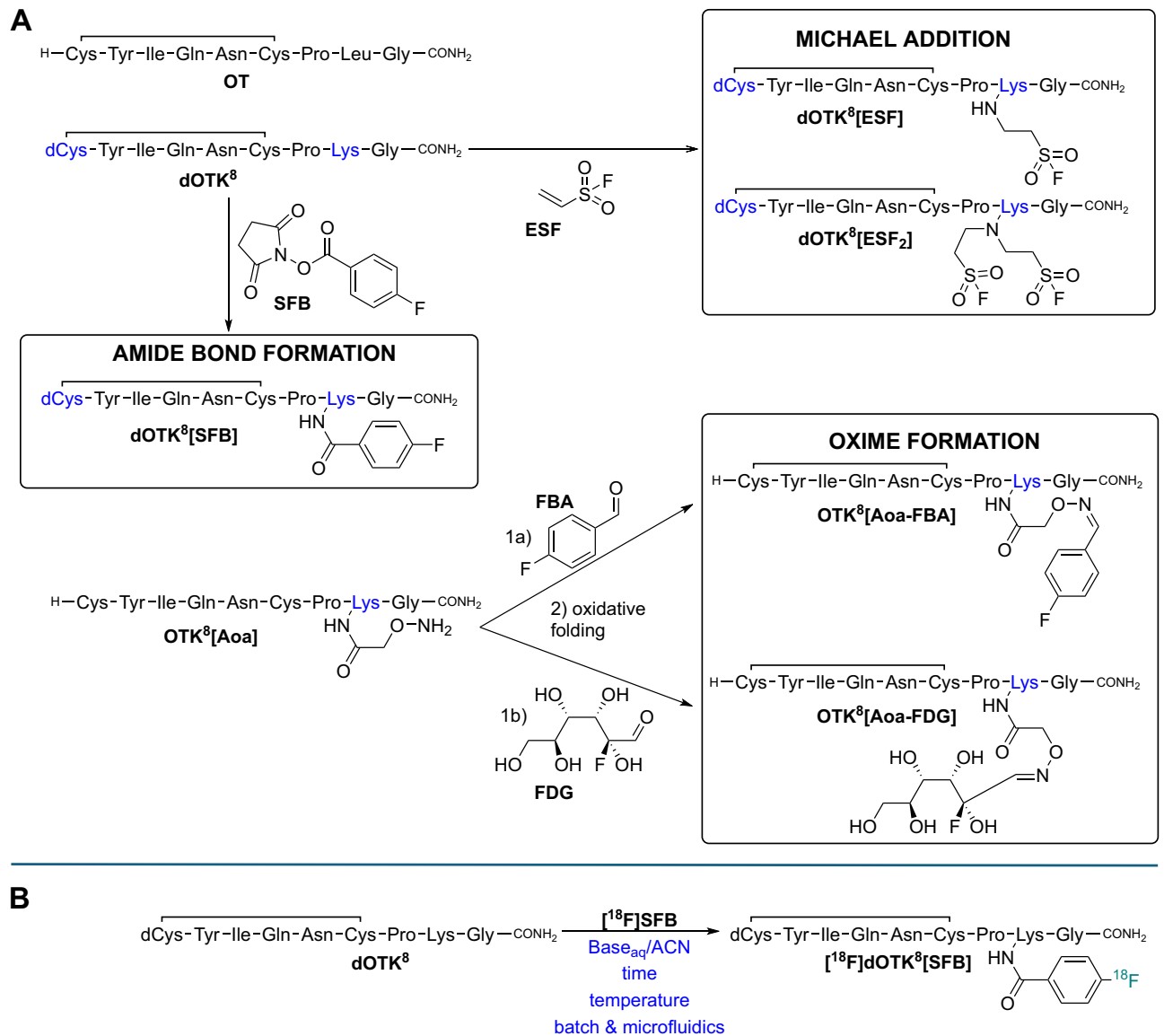

**Fig. 1 | Chemical and radiochemical analogues studied. A** Chemical structures of studied OT analogues and respective synthetic route. Aoa = aminooxy acetic acid; SFB = N-succinimidyl-4-fluorobenzoate; FDG = fluorodeoxyglucose (acyclic aldehyde representation); ESF = ethenesulfonylfloride; FBA = 4-Fluorobenzaldehyde. **B** Reaction scheme for the radioconjugation of **dOTK8** using [18F]**SFB**. Aspects that were varied during optimisation are highlighted under the reaction arrow.

As observed for OT, all candidates retained potency at $V_{1a}R$ and $V_{1b}R$, supporting their OT-like pharmacological character.

**Justified selection of lead OT analogue for radiolabelling.** Considering the biophysical data collected for the candidates, we selected **dOTK8[SFB]** as the most OT-like ligand and preferred lead for the radiopharmaceutical development (Fig. 2D). **OTK8[Aoa-FDG]** was the second-best candidate; however, like **OTK8[Aoa-FBA]**, it raised concerns due to the need for optimisation of radiolabelling *via* oxime ligation and its limited serum stability. **dOTK8[ESF]** also looked promising; however, it was reported that the [18F]**ESF** moiety can defluorinate in serum[35], which is not ideal for PET imaging. Nevertheless, we also tested the feasibility of its radiobioconjugation on **OT** and **dOTK8**, given the innovative character of such a new group (*SI-7*).

**18F radiolabelling, process optimisation, and radiopharmaceutical production**
Radioconjugation of **dOTK8** (Fig. 1B) was tested using a solution of [18F]**SFB** in 1 mL of ACN and 1 mg of the peptide dissolved in 1 mL of 0.05 M

$Na_2HPO_4$ at pH of 8.5. The radioconjugation reaction was tested both in batch and flow microfluidic reactor environments. Direct radiolabelling of **OT** at its N-terminus using the same [18F]**SFB** conditions was performed as a synthetic control. However, due to the known sensitivity of the N-terminus to modification, which reduces both receptor binding and activity[36], this construct was not pursued further for biological evaluation.

**Vessel radiolabelling approach.** The radioconjugation reaction was tested at 100 μL scale, adding 50 μL of [18F]**SFB** to the same volume of a 1 mg/mL solution of either OT or **dOTK8** in reactions vials heated at 25 °C, 50 °C, or 70 °C. The reactions were run for 10 min or 20 min and quenched by the addition of 100 μL of ACN. Radiochemical yields (RCY)[37] of these reactions were determined by radio-HPLC, based on the initial [18F]**SFB** activity and calculating the peak area of [18F]**dOTK8[SFB]** relative to the areas of all the other radioactive peaks (Fig. 3A).

Radiolabelling of **dOTK8** performed better than **OT** across all three temperatures and reaction times. A longer reaction time of 20 min provided higher RCY at 25 °C and 50 °C, with some product degradation observed at

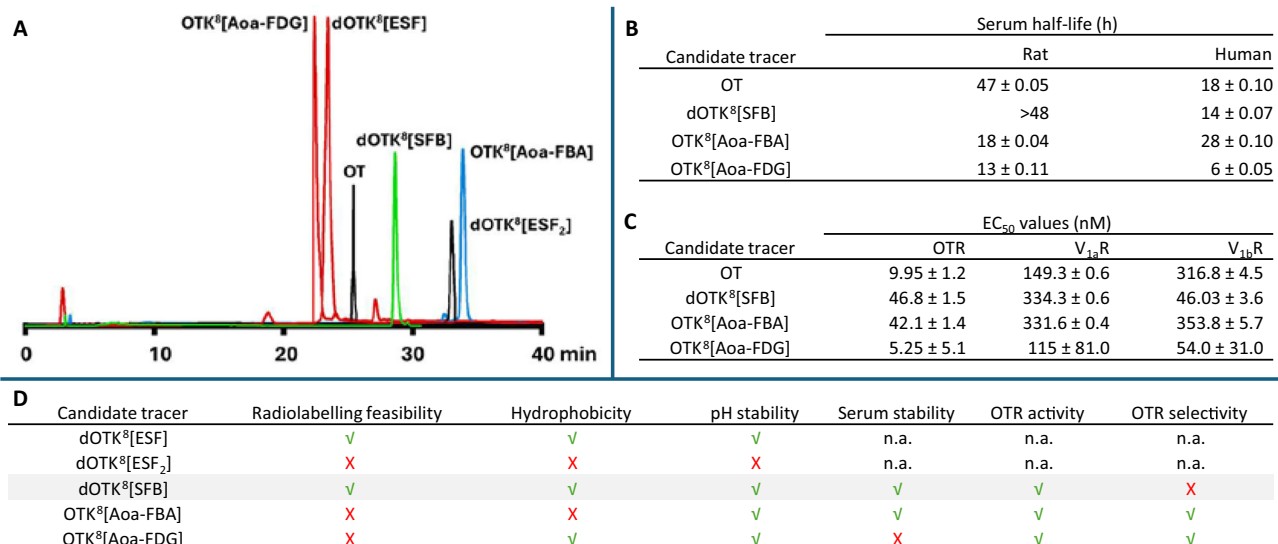

**Fig. 2 | Pharmacological characterisation of OT analogues. A** OT analogues hydrophobicity test assessed by HPLC (HPLC Column: Vydac 218TP52, $C_{18}$, 10 μm, 22 × 250 mm, 300 Å; gradient method: 0–40% B, 1%/min, 0.3 mL/min; solvent A: 0.05% TFA in $H_2O$, solvent B: 0.043% TFA in 90% ACN). **B** Metabolic stability of OT analogues in rat and human serum. Each time point was analysed in three independent experiments ($n$ = 3 technical triplicates, mean ± SEM). **C** Potencies ($EC_{50}$) values for OT and analogues at the human OTR, V1aR, and V1bR, determined by a FLIPR calcium mobilisation assay ($n$ = 3 technical triplicates, mean ± SEM). **D** Synoptic table of features tested for the analogues in this work and their similarity assessment with native OT. Desired criteria: anticipated optimal radioconjugation in < 1 h and activity yield > 15%; similar hydrophobicity with HPLC Rt ± 20% from OT; buffer $t_{1/2}$ > 30 min at pH 7; metabolic $t_{1/2}$ > 10 h in rat and human serum; $EC_{50}$ at OTR < 50 nM; $EC_{50}$ at V1aR and V1bR < 5×$EC_{50}$(OTR) (n.a.: not assessed).

70 °C, and the degradation product eluting closely to the desired product for both substrates (*SI-8, SI-9*). The best condition was at 50 °C for 20 min, yielding [$^{18}$F]**dOTK$^8$**[**SFB**] with an 86% RCY.

**Microfluidic radiolabelling approach, HPLC purification, and formulation.** Flow microfluidic systems provide an environment where multiple reactions can be run sequentially with high repeatability and low consumption of chemicals; this feature is important when optimising reaction conditions with radioactive or expensive reagents[38,39]. In addition, the ease of scaling up by simply increasing the volume of reagents is attractive for moving to production scale. This is typically not easily achieved in a vessel approach, where dedicated re-optimisation is required for scale-up.

**Small-scale optimisation.** The same radiolabelling reactions were tested in an Advion NanoTek microfluidic system, using a 1.15 m PEEK tube reactor with an inner diameter (I.D.) of 0.25 mm (i.e. 56 μL volume) and delivering reaction boluses of 20–40 μL. Different conditions were tested by varying overall flow rates, reactant flow ratios, and temperature. Temperature turned out to be particularly important, with the 50–70 °C range having the best RCY. Higher temperatures resulted in increasing amounts the degradation product already identified in vessel reactions (*SI-10, SI-11*). 50 °C was selected as the preferred temperature in order to minimise degradation when switching to the larger reaction boluses in the production stage (Fig. 3B)[40]. When optimising reactant flow ratios, peptide solutions were used at concentrations of 1 mg/mL and usable reaction yields were obtained by merging isovolumetric amounts of [$^{18}$F]**SFB** solutions, thus realising a 0.5 mg/mL peptide concentration (~0.5 mM) in the flow reactor. Using higher volume ratios of peptide provided marginally higher yields and did not justify the increased expense of chemicals. When testing flow rates of 5 and 50 μL/min for each reagent channel, better RCYs were observed at lower flow rates for both peptides. At 50 °C, a RCY of 65% and 20% were obtained with a 5 μL/min flow (i.e. 10 μL/min cumulated, 5.6 min residence time) with **dOTK$^8$** and **OT**, respectively. However, using a 10 μL/min flow rate (i.e. 20 μL/min

cumulated, 2.8 min residence time) for our key target **dOTK$^8$** gave a 47% yield (Fig. 3C).

Given that larger boluses of 100–200 μL were required to obtain sufficient radioactivity for animal imaging, the faster flow rate of 10 μL/min (i.e. 20 μL/min cumulated) was selected for the production stage. Using a slower flow rate of 5 μL/min would decrease radioactivity recovery by 12% due to an additional 20 min of decay (i.e. 200 μL bolus level), thus nearly abating the 18% increase of RCY. A faster flow rate also accelerates the overall production process, providing more time for imaging and multi-batch productions, if necessary[41].

**Semi-preparative $C_{18}$-RP-HPLC purification of radiotracer.** Since the reaction mixture of the conjugation reaction comprised four main radioactive components due to incomplete conjugation and hydrolysis, namely [$^{18}$F]**SFB**, [$^{18}$F]**dOTK$^8$**[**SFB**], [$^{18}$F]**FBA** and an unknown species, a semi-preparative RP-HPLC purification step was required to afford the desired radiotracer in sufficient purity for PET imaging (>95%). Screening of different semi-preparative (semi-prep) elution conditions on a monolith semi-prep column (Merck Chromolith, 150 × 10 mm, 13 nm pores) was performed by analysing the retention times of non-radioactive standards and using varying ratios of $H_2O$ and ACN (with 0.05% of TFA, 2.5 mL/min flow rate). No baseline separation was possible with >30% of ACN, while at 23% ACN, the most hydrophobic analyte (**SFB**) was retained for >25 min, enabling the separation of the desired product. With two back-to-back productions of [$^{18}$F]**dOTK$^8$**[**SFB**] in mind, the 25/75 = ACN/$H_2O$ (with 0.05% of TFA) mixture provided good results, yielding a good separation in a short run time of 20 min. These conditions were tested by loading a 500 μL injection loop with a production-level run (see below). An example of the semi-prep radio-chromatogram is provided in Fig. 4A, also indicating the relation of the collected fractions with the analytical chromatogram of the mixture (Fig. 4B), and noticing an inversion of $R_t$ for fractions F2 and F3 between the two chromatographic system. Prefilling the injection loop with water before loading the reaction mixture was crucial for the success of the purification, as small aliquots of organic solvents (e.g. from a previous

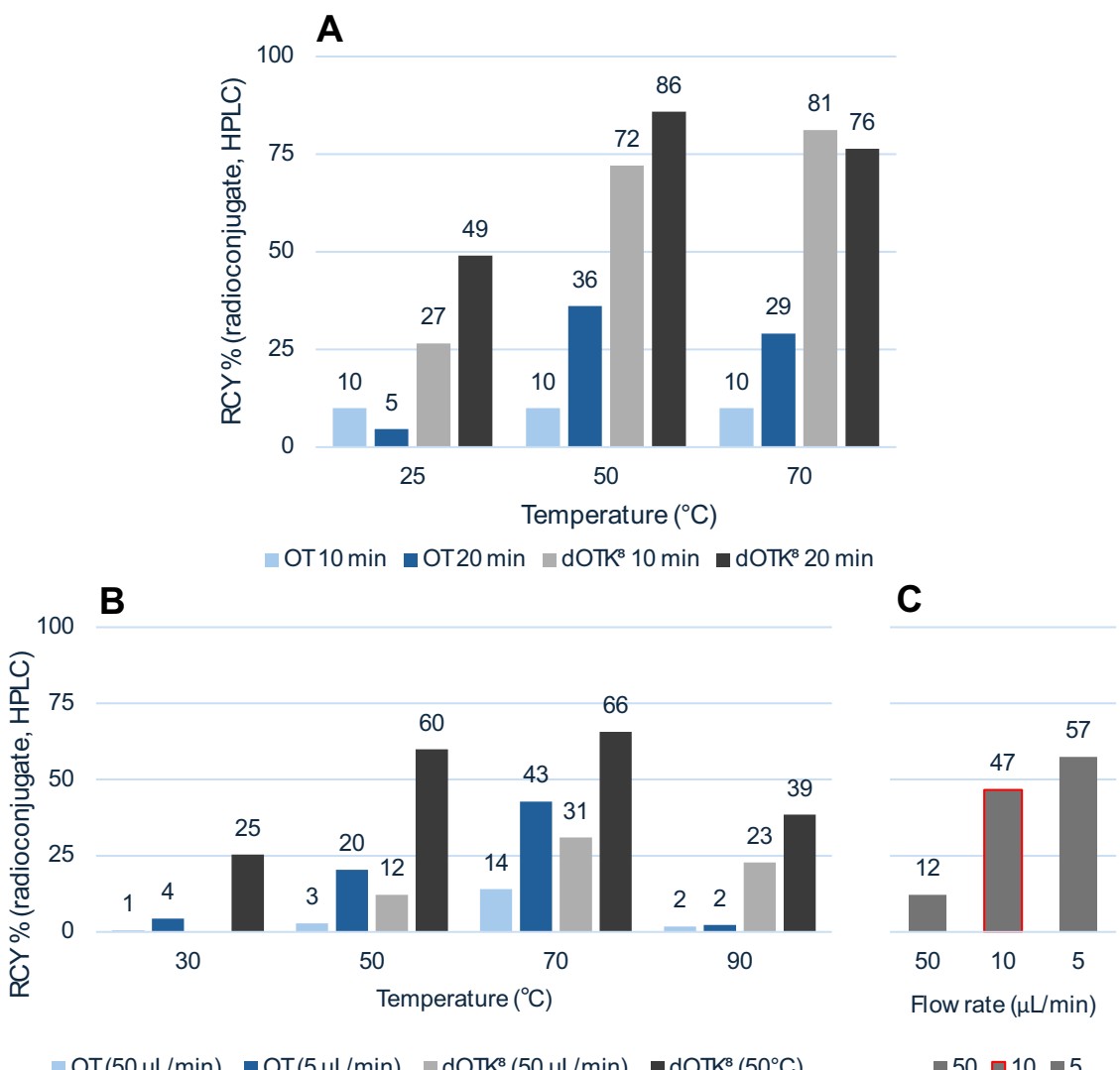

**Fig. 3 | Optimisation of radiolabelling of OT and dOTK[8] precursors with [[18]F] SFB. A** Radiochemical yields (RCY) for vessel radioconjugation of **OT** and **dOTK[8]** with [[18]F]**SFB** evaluated by Radio-HPLC ($n = 1$). **B** Comparison of radiochemical yields for the radioconjugation of **OT** and **dOTK[8]** with [[18]F]**SFB**, under different microfluidic conditions ($n = 1$). **C** Comparison of radiochemical yields for the radioconjugation of **dOTK[8]** with [[18]F]**SFB**, under different flow rates at 50 °C ($n = 1$).

HPLC run) increased the break-through of the mixture with the injection void volume.

**Formulation of the finished product**. The product collected in semi-prep HPLC previously optimised (Fraction **F2**, 2.5–5 mL) was trapped in a preconditioned $C_{18}$ SepPak cartridge without pre-dilution with $H_2O$; this aspect considerably simplifies the process and hardware required, traditionally needing addition of several volumes of water to ensure complete trapping in the $C_{18}$ matrix. After trapping, $H_2O$ rinsing, and $N_2$ drying were performed, and the product was eluted with 0.75 mL of EtOH, followed by EtOH evaporation under $N_2$ stream at 60 °C in a plastic (cycloolefin copolymer, COC) vial, to minimise product losses compared to glass vials. This simplified formulation process resulted in a recovery of $62 \pm 7\%$ of the collected radioactive product, with $14 \pm 8\%$ lost due to insufficient retention during cartridge loading (i.e., breakthrough). A small liquid residue (<50 µL) typically remained, likely to be residual $H_2O$ since no organic solvents were detected by GC-MS. Dilution with isotonic saline was kept at a minimum, yielding [[18]F]**dOTK[8][SFB]** with 2.7 MBq/µL radioactive concentration on average.

**Full production process and quality control**. The production stage for [[18]F]**dOTK[8][SFB]** was carried out using a custom-modified Advion

Nanotek system[42], using 100-200 µL volumes for each reactant channel. The following preferred reaction conditions were adopted: a solution of [[18]F]**SFB** (>15 Gbq in 1.5 mL of $CH_3CN$) and a solution of **dOTK[8]** (0.5 mg in 0.75 mL of 0.05 M $Na_2HPO_4$, pH 8.5) were delivered at 10 µL/ min each in the 56 µL PEEK microreactor kept at 50 °C. This setup resulted in an RCY for the radioconjugation steps of $46 \pm 8\%$, estimated using 10 µL boluses of reactants before or after production runs (i.e. "dummy" runs method).

The full production process—including microfluidic radio-conjugation, HPLC purification, and formulation—was performed back-to-back[43], yielding two separate product batches from the same starting radioactivity and peptide precursor. Following the delivery of [[18]F]**SFB** to the microfluidic system, 30 min were required to load the storage loops with reagents and prepare the synthesiser. Once initiated, the production run took 60 min to complete. This process can be started at any time, provided a sufficient volume of radioactivity is available. Therefore, the radioactivity output of the final product was dependent on several parameters, such as: starting amount of [[18]F]**SFB**, volume of the radioactive bolus used, and being 1st or 2nd production batch (from the same starting radioactivity). On average, 149 MBq for each 100 µL of [[18]F]**SFB** was obtained, corresponding to 169 MBq for the 1st batch ($N = 9$) and 104 MBq for the 2nd batch ($N = 4$). RCY was assessed using two approaches. First, based on starting [[18]F]**SFB**

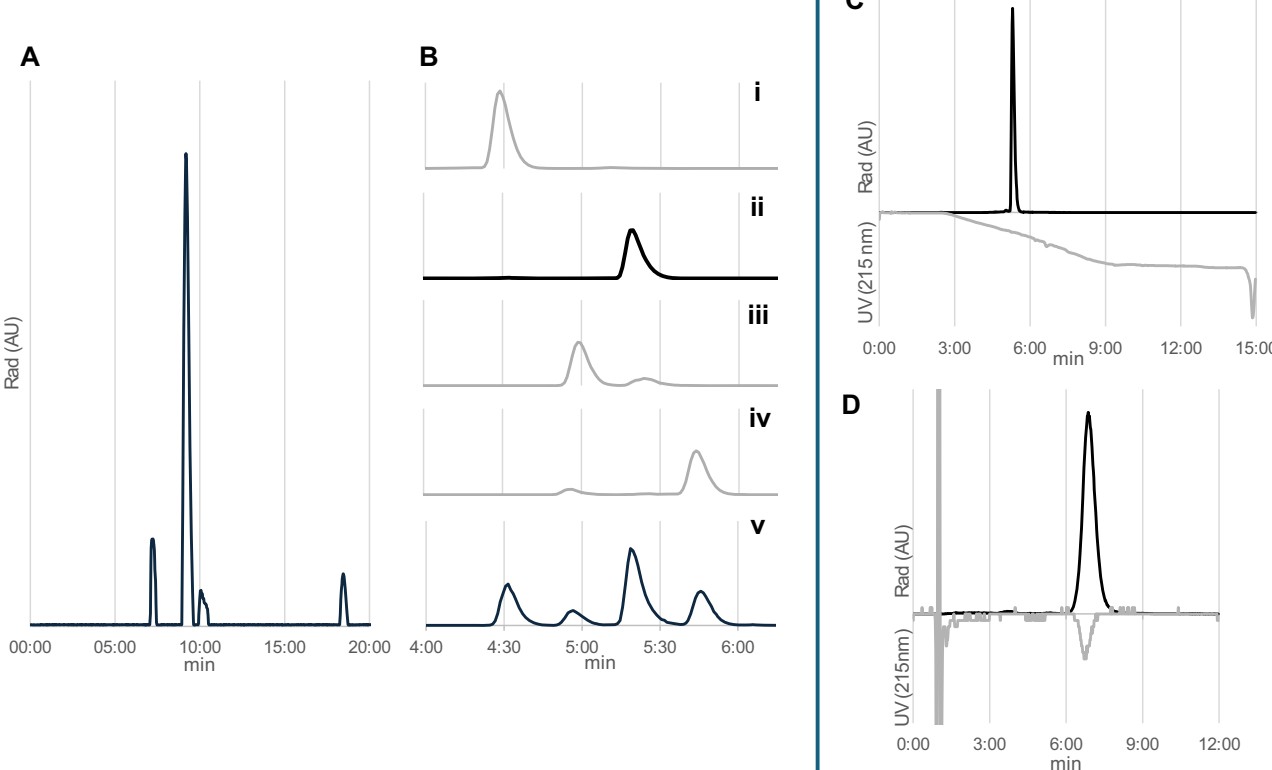

**Fig. 4 | Purification and analysis of the final product. A** Radio-HPLC profiles for the conjugation reaction mixture injected in the semi-prep system (Chromolith RP semi-prep column (150 × 10 mm, 13 nm pores), using a 25/75 = ACN/$H_2O$ with 0.05% TFA (v/v) at a 2.5 mL/min flow rate), highlighting the fraction collected (F1-F4). **B** Expanded portion of radiochromatogram (method B, 4:00–6:30 min) analysing the 4 fractions collected in the semi-prep run (i-iv), compared with the unpurified mixture analysed in the same conditions (v); F2 (ii) revealed to be the desired product. **C** HPLC profiles for quality control of final [$^{18}$F]**dOTK$^8$[SFB]** product using gradient conditions (method B) to quantify radiochemical purity. **D** HPLC profiles for quality control of final [$^{18}$F]**dOTK$^8$[SFB]** product using isocratic conditions (method C) to quantify molar activity.

using the 'dummy' method[44], resulting in a value of 25 ± 7%. Secondly, RCY was also calculated based on the starting amount and volume of [$^{18}$F]**SFB**, as measured by the Synthra radiodetector, which resulted in a lower RCY of 19 ± 8%. This small discrepancy may be due to an overestimation of the starting [$^{18}$F]**SFB** caused by detector spillover signal (i.e. radioactive shine) in the Synthra system. Nevertheless, the agreement between these two independent assessment methods supports the reliability of the reported RCY values and demonstrates that very little radioactivity was lost in the microfluidic system. This case is substantially different from radiofluorination reactions performed in glass microfluidic chips, whereas varying amounts of radioactivity are retained in the reactor and not accounted for in the final product[45].

The HPLC analysis of [$^{18}$F]**dOTK$^8$[SFB]** was performed with a sequential injection protocol, whereas the product was injected twice using two different elution methods on the same column. Such an approach was preferred, as the gradient method was previously extensively used to evaluate the various radioactive by-products occurring in the conjugation process, while the isocratic method provided a stable chromatographic baseline at 215 nm, which allowed integration of the small UV product peak and accurate calculation of molar activity ($A_m$), that was 191 ± 97 MBq/nmol at EOS.

To enable production in laboratories without access to a microfluidic system, the synthesis of [$^{18}$F]**dOTK8[SFB]** was successfully adapted to a conventional vessel-based method (i.e., using the Flexlab synthesiser), yielding comparable radiochemical purity and yield.

## Biodistribution study

Pharmacokinetics (PK) and tissue biodistribution were determined after intravenous injection of [$^{18}$F]**dOTK$^8$[SFB]** in healthy rats. Blood samples were collected at regular intervals over a 1-h period, and radioactivity was measured using gamma counting. Subsequently, animals were humanely euthanized, and all major organs and tissues were collected to determine tracer biodistribution. The blood and plasma PK profile of our tracer revealed peak detection at ~1.5 min after tail vein injection, followed by rapid clearance within 5–6 min, consistent with reported PK profiles for OT and other OT peptide ligands (Fig. 5A)[27,46]. The biodistribution study (Fig. 5B) revealed significant excretion of the tracer *via* urine (36–40%ID/g), followed by high signals in the kidneys (0.9–1.1%ID/g), small intestine (0.4–0.6%ID/g) and liver (0.4–0.5%ID/g), consistent with data from another PET OT ligand[20]. Organs of interest for OTR presence and OT biology, such as mammary pads, testes, and thyroid, showed no abnormal biodistribution. Relatively low tracer uptake was observed in the collected brain regions, although a notably higher value was recorded in the posterior section. Importantly, the low uptake in bone confirmed the stability of the designed radiolabelling position, as defluorination would result in high bone uptake.

## Micro-PET/CT animal imaging

**Pilot PET/CT imaging study.** Preliminary micro-PET/CT imaging was performed on three healthy male rats (T1-T3) by injecting 30–60 MBq of [$^{18}$F]**dOTK$^8$[SFB]** *via* tail vein (~0.2–0.6 nmol of tracer from $A_m$ calculation) as a single bolus in three independent experiments. Dynamic scanning was performed for 60 min on the top-half of the body, and regions of interest (ROIs) were delineated around the pituitary glands and mammary fat pads, as they exhibited noticeable tracer uptake (Fig. 5C and Supplemntary Video 1). Both these ROIs are rich in OTR: the pituitary gland is the key organ involved in the release of OT from the brain into the bloodstream, while OTR is critical for milk let-down in the

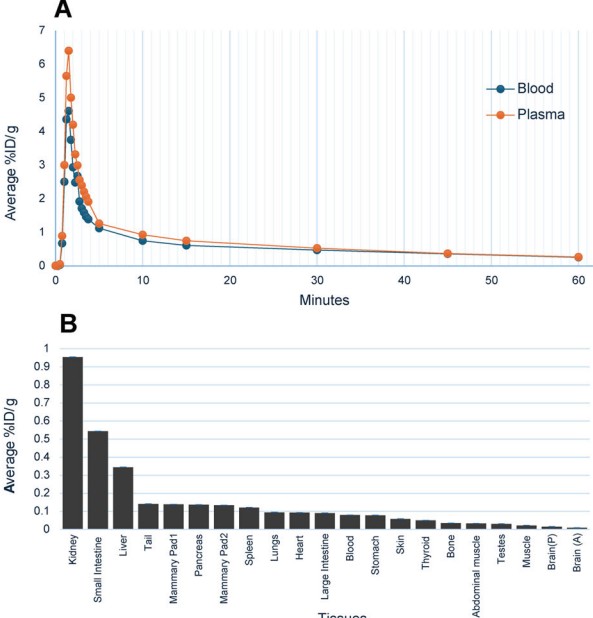

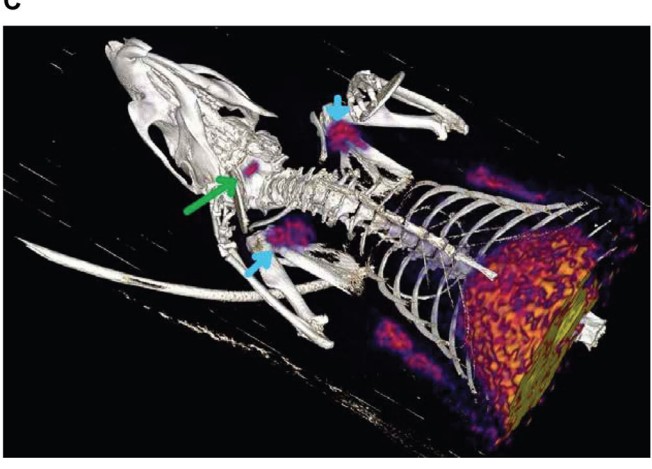

**Fig. 5 | Initial PET biodistribution assessment. A** Radiotracer concentration (shown as %ID/g) in blood and plasma samples collected over 60 min (average of $n = 5$). **B** Biodistribution data at 60 min post injection (shown as % injected dose per gram of tissue weight, %ID/g) ($n = 3$, ± SEM, urine data excluded for clarity). **C** PET/ CT 3D fusion render highlighting regions of specific uptake (green arrow: pituitary gland; blue arrows: mammary pads); in the lower region of the field of view it is possible to notice two more sets of mammary pads and substantial liver and intestines uptake.

mammary pads[47]. No tracer uptake was observed in the brain; however, it is worth noticing that although the pituitary gland is considered part of the brain, it is not protected by the blood-brain barrier (BBB)[48].

Tracer target specificity was demonstrated in a separate pre-blocking experiment (T4), where 100 µg (~5 times the amount of OT used in rat studies[49]) of non-radioactive isotopologue [$^{19}$F]dOTK$^8$[SFB] was administered 20 min prior to injection of radioactive [$^{18}$F]dOTK$^8$[SFB]. Uptake in the ROIs previously identified (pituitary gland and mammary pads) was significantly suppressed, to a degree where it was difficult to visually locate the ROIs. Anatomical referencing was then used to locate the ROIs, and tracer specificity was confirmed by comparing the time activity curves (TAC) of the three initial pilot imaging experiments with this pre-blocking experiment (Fig. 6). The TAC of the radiotracer in the heart region (i.e. blood pool) was used as control, showing no noticeable suppression effects.

The OT-OTR complex is known to internalise following receptor activation[50]. Displacement experiments were conducted to verify the anticipated irreversible trapping of our OT-like radiotracer through tracer-OTR internalisation. Specifically, 100 µg of non-radioactive [$^{19}$F] dOTK$^8$[SFB] or OT was administered 60 min after radiotracer injection. No significant reduction of radiotracer uptake was observed upon any ligand administration, thus confirming that [$^{18}$F]dOTK$^8$[SFB] internalised as expected (*SI-15*).

**Grouped PET/CT imaging study.** Considering the promising findings of the pilot experiments, we carried out micro-PET/CT studies in healthy male rats distributed in three study groups. In the first group, a 60 min dynamic scan was performed by injecting only the radiotracer, spiked with non-radioactive isotopologue to reach a total of 1 nmol injected (i.e. 'baseline', $n = 13$). In the second experiment type (i.e. 'blocking'), a larger amount (100 µg) of non-radioactive isotopologue ($n = 2$) or OT ($n = 3$) was administered 30 min before the radiotracer. Image analysis was performed as previously by drawing ROIs in pituitary glands and mammary pads; however, in this case, a comparison of TACs showed a reduced blocking effect for mammary pads and little apparent blocking for pituitary glands (Fig. 7A–D). As expected, no significant difference was noted between the left and right mammary pads.

**Kinetic analysis.** We performed kinetic modelling based on Patlak graphical analysis to verify tracer specificity. This approach returns a single macro-parameter of interest, $K_i$ [mL/cm$^3$/min] (i.e. the irreversible uptake constant in tissue) and it requires two conditions to be satisfied: (i) the presence of an irreversible system compartment and (ii) equilibration, after a certain time, of all the reversible compartments[51]. The Patlak plot usually requires tracer plasma concentration as reference region, but it can be adapted to use a compartment void of specific receptors instead (*SI, Section 9*). In this study, the mammary pads and pituitary gland were used as the target compartment, while the heart content (i.e. blood pool) was used as the reference compartment. This reference ROI was delineated manually but fixing its 3D shape and placing such volume within the heart cavity. The Patlak equation used in this study is given by:

$$\frac{C_{ROI}(t)}{C_{heart}(t)} = K_i^{ref} \frac{\int_0^t C_{heart}(\tau)d\tau}{C_{heart}(t)} + q$$

where $K_i^{ref}$ is the parameter of interest, estimated after a time $t^\cdot$ after which the plot becomes linear. Moreover, $C_{heart}$ is the concentration of the tracer in the reference region and $C_{ROI}$ is the concentration of the tracer in the region of interest, either mammary pad or pituitary gland.

From this analysis, it was possible to notice that significant differences ($p < 0.05$) exist between the baseline studies and both OT ($p = 0.0383$) and [$^{19}$F]dOTK$^8$[SFB] ($p = 0.0206$) blocking experiments in mammary pads regions (Fig. 7E). In the pituitary glands (Fig. 7F), both blocking experiments gave a lower average value for $K_i^{ref}$, but such difference was not significant ($p = 0.8191$ and $0.7409$ for OT and [$^{19}$F]dOTK$^8$[SFB], respectively). In addition, no significant differences were noticed in the mammary regions between the two blocking molecules ($p = 0.9793$ in pituitary, $p = 0.8139$ for mammary pads average), thus substantiating that [$^{19}$F]dOTK$^8$[SFB] is an appropriate analogue of OT.

## Conclusions

This study focused on developing an OT-like radiotracer to enable PET imaging of the OT/OTR system, thereby expanding the molecular toolbox for investigating this ancient and multifaceted GPCR[2]. Following

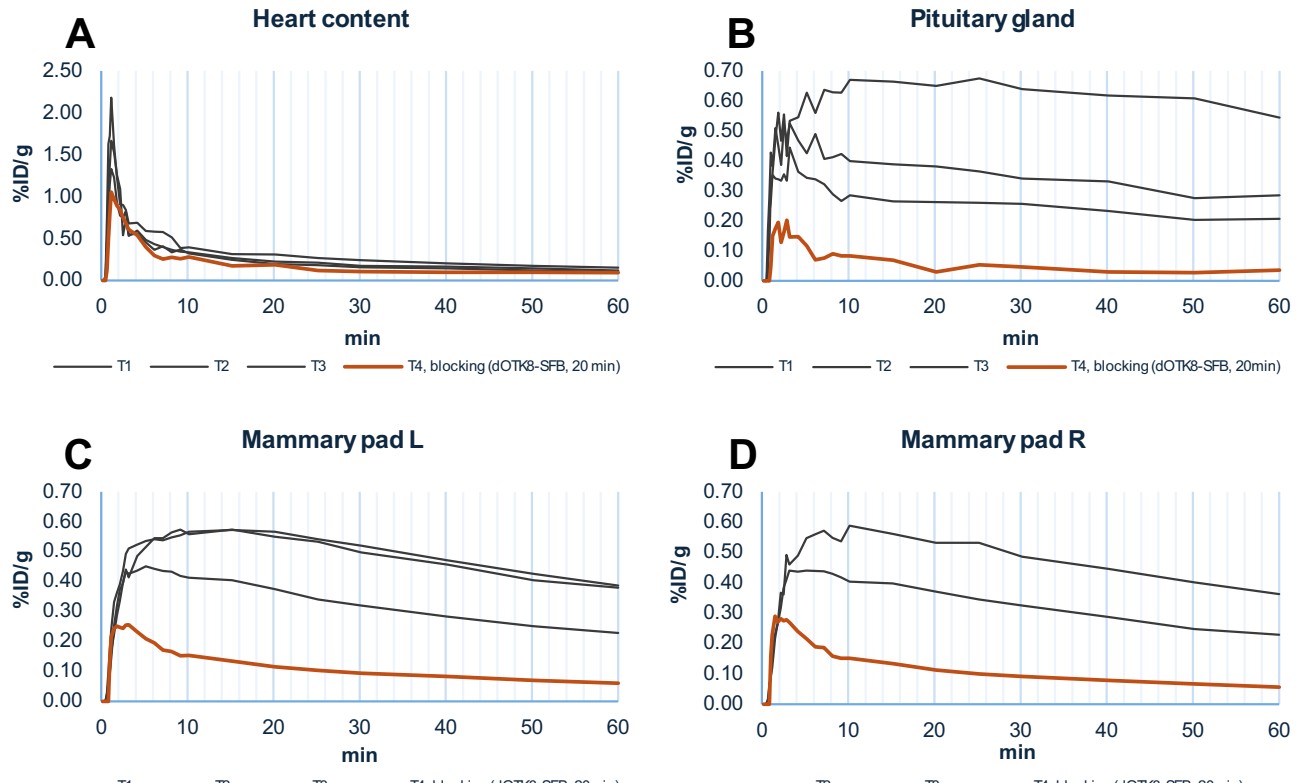

**Fig. 6 | Pilot PET imaging study results.** TACs for each of the pilot experiments (%ID/g vs min), compared with data from 20 min prior pre-blocking using non-radioactive isotopologue, in the heart (control, **A**), pituitary gland (**B**) and left (**C**) and right (**D**) mammary pads (it was not possible to locate the right mammary pad of T1 subject).

conventional peptide labelling strategies[52], we designed and tested five OT analogues, and finally selected [$^{18}$F]**dOTK$^8$[SFB]** as the best lead for PET imaging based on physicochemical, pharmacological, radiochemical, and practical considerations.

Radiopharmaceutical production was optimised using a microfluidic approach, achieving a 25% RCY with >95% purity and an average $A_m$ of 191 MBq/nmol. It is worth noting that such $A_m$ value is in the higher range when compared to the amount of radioactivity obtained for each batch (<200 MBq), thus strengthening the case for the benefits of a microfluidic environment[53]. Moreover, microfluidic system allows fast process optimisation and seamless translation to preclinical dose production levels with high yields, reliability, and flexibility in manufacturing on-demand radiotracer batches. In addition, we demonstrated the importance of identifying an optimal flow rate for microfluidic production that balances RCY with overall process duration. We prioritised flow rates that offered slightly lower RCY in exchange for shorter production times, thereby improving reliability of activity yields and facilitating better management of imaging experiments. A more conventional vessel-based production method has also been reported, yielding similar efficiencies and quality.

In PET imaging experiments in rats, [$^{18}$F]**dOTK$^8$[SFB]** displayed regions of uptake as expected of OT, in particular highlighting an irreversible uptake in the pituitary gland and mammary pads, known to be rich in OTR. Blocking experiments with both OT and the non-radioactive isotopologue (100 μg bolus for both) yielded a reduction of the Patlak $K_i^{ref}$ in mammary pads and the pituitary gland, with only the former tissue reporting significance ($p < 0.05$, Fig. 7E). The loss of significant blocking in the pituitary gland compared to the single pilot experiment could be attributed to the different imaging conditions used in the successive grouped imaging campaign, in particular for what regards fixed-mass tracer injection (0.3 nmol for pilot *vs* 1 nmol for campaign) and time gap after competitor injection (20 min for pilot *vs* 30 min for campaign). There is evidence that various tissues have different times of OTR resensitization[50], and this opens the possibility that

pituitary gland, the key organ managing the trafficking of OT, has a faster cycle of re-expression of such receptor. For this region, using a time gap of 30 min would allow resensitization to happen and tracer to find new receptors expressed and ready to bind; on the other hand, when using a time gap of 20 min, the tracer will find significantly less new receptors expressed in the region (thus showing a successful blocking), but the fast metabolism of OT analogues (3–10 min)[54] will not allow the parent tracer to be present when the receptor is re-expressed in the pituitary gland (*SI-18*). However, additional studies with variable injection time gaps are required to confirm this hypothesis, and this currently represents a limitation of our work. However, if confirmed, such observation may induce consideration when using such imaging tools to assess the potency of novel OTR-targeted treatments in human studies, whereas high $A_m$ and different times of pre-blocking might need to be trialled.

No brain uptake of our tracer was observed during PET imaging in our animal model; however, the study was not aimed at developing a brain-penetrant tracer nor studying brain uptake in rats, but at developing and validating an OT-like PET tracer to enable the first and more relevant human studies. Furthermore, rodents have a very different nasal anatomy compared to humans; hence, we did not perform any PET imaging following intranasal administration. However, future human PET imaging studies are planned to investigate these aspects.

In summary, [$^{18}$F]**dOTK$^8$[SFB]** represents the first successful example of an OT-like PET tracer covalently radiolabelled with a positron emitter, with a suitable imaging profile in preclinical PET. In particular, we demonstrated, for the first time with in vivo imaging, its specificity and selectivity for OTR-rich tissue (e.g. mammary pads). Therefore, the progression of this molecule as a radiopharmaceutical for imaging the OT system in humans is warranted, to better understand the role of this important biochemical pathway in several disease and treatment strategies. This may advance personalised medicine approaches to OT-based therapies in humans and improve therapeutic targets and responses.

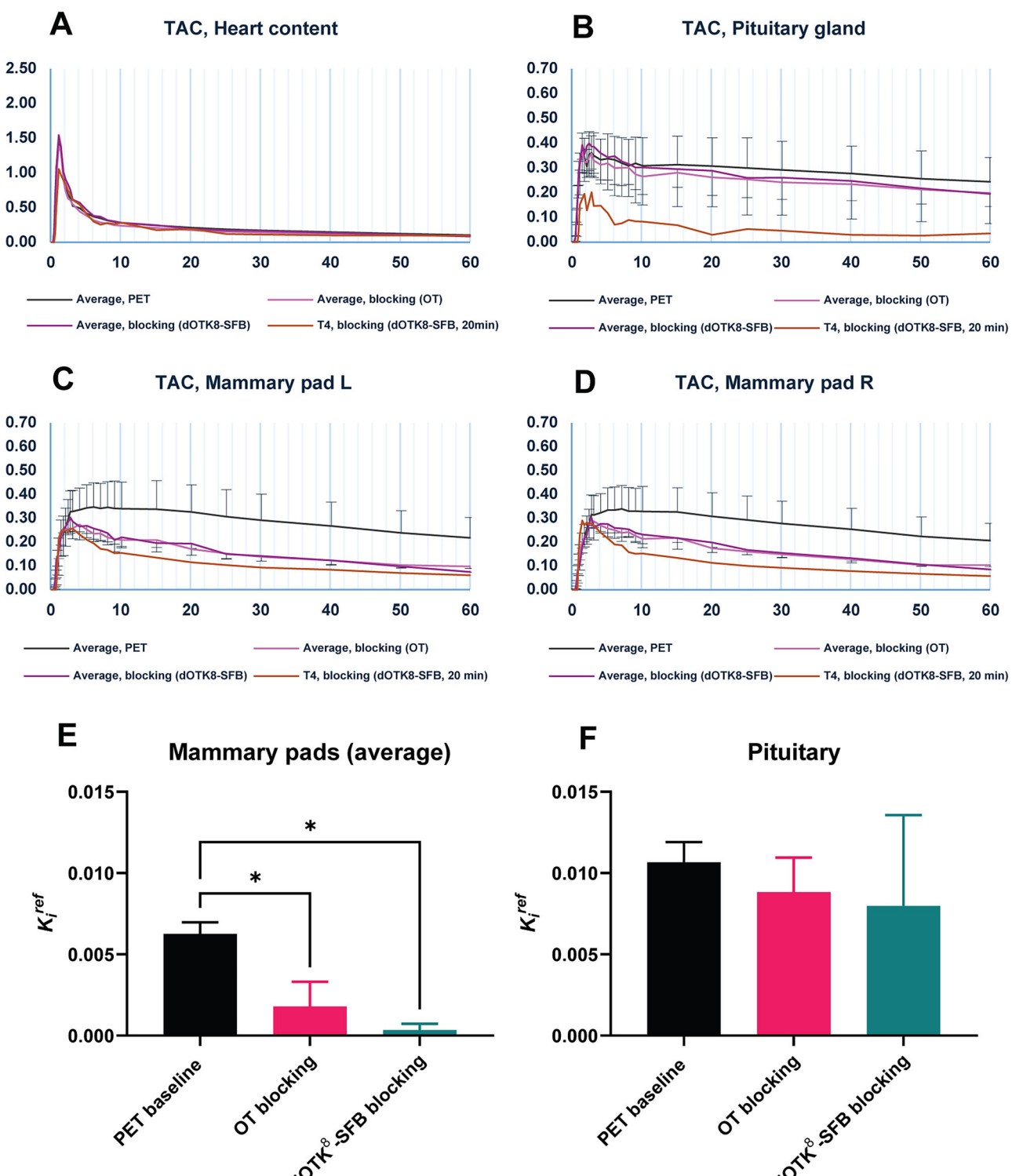

**Fig. 7 | Grouped PET imaging study and Patlak plot results. A**-**D** TAC average curves (%ID/g vs min) for the three groups of animal experiments in four ROIs, compared with the data from the pilot blocking experiment (T4); positive error bar for baseline PET, negative error bar for blocking experiments, no error bar for heart content data (error bars: STD). **E, F** Patlak $K_i^{ref}$ values of two tissues of interest (left and right mammary pads have been averaged) for the three groups (error bar: SEM); $p$ values calculated with one-way ANOVA, with significant differences marked (*: $p < 0.05$); the parameter reported is the average of choosing the last 4, 5 and 6 time points of the Patlak plot.

## Experimental section
### Chemistry and pharmacology
**Materials**. All solvents and reagents were used as supplied. Fmoc amino acid derivates (Fmoc-Gly)-OH, (Fmoc-Lys(Mtt))-OH, (Fmoc-Pro)-OH, (Fmoc-Cys(Trt))-OH, (Fmoc-Asn(Trt))-OH, (Fmoc-Gln(Trt))-OH,

(Fmoc-Ile)-OH, (Fmoc-Ile)-OH and Fmoc-protected S-Trityl-3-mercaptopropionic acid ((Fmoc-dCys(Trt))-OH) were purchased from Iris Biotech. The Rink amide resin (Polystyrene AM RAM; 0.69 mmol/g) was obtained from RAPP Polymere. Triisopropylsilane (TIPS), 1,2-ethae-dithiol (EDT), ammonium bicarbonate ($NH_4HCO_3$), ethensulfonylfloride

(ESF), 2-Fluoro-2-deoxy-D-glucose (FDG), 4-fluorobenzaldehyde (FBA), 4-fluorobenzoic acid (FBAc), N-hydroxy-succinimide (NHS) were obtained from Merck. The coupling reagent O-(7-azabenzotriazol-1-yl)-N,N,N′,N′-tetramethyluronium hexafluorophosphate (HATU) and bis-Boc-aminooxy acetic acid (Bis-Aoa) were purchased from Chem-Impex International; N,N-diisopropylethaylamin (DIPEA) and N,N′-dicyclohexylcarbodiimide (DCC) fro Auspep; dimehtylformamide (DMF) and diethyl ether from RCI Labscan Limited; trifluoroacetic acid (TFA), piperidine and methanol (MeOH) from Chem-Supply; dichloromethane (DCM) and acetonitrile (ACN) from Merck. N-succinimidyl-4-fluorobenzoate (SFB) was synthesised with minor modifications from literature procedures[55].

**Solid-phase peptide synthesis (SPPS).** Peptides were synthesised manually on Rink-amide resin (0.125 mmol scale). The resin was washed with DMF and left swelling in DMF overnight. Fmoc-deprotection was performed using 30% piperidine in DMF ($2 \times 30$ s) and washed with DMF again. The incoming amino acid (4 eq relative to the resin loading) was activated with HATU:DIPEA (1:1:1) in DMF and added to the resin. The coupling reaction was stopped after 15 min by washing the resin with DMF, and the synthesis cycle of Fmoc-deprotection and amino acid coupling was repeated upon completion of the peptide sequence. After the peptide sequence was fully assembled, the resin was washed with DCM and MeOH, followed by peptide cleavage from the resin and removal of protection groups by treatment with TFA:TIPS:EDT:H$_2$O (90:2.5:2.5:5) at room temperature (25 °C, r.t.) for 80 min. TFA was evaporated under a stream of nitrogen, and the peptide precipitated with ice-cold diethyl ether. The precipitate was filtered off and dissolved in 50% ACN/H$_2$O + 0.1% TFA and lyophilised overnight. The crude products were analysed via LC-MS (SI-19). The linear OT-like peptides were dissolved in a minimal amount of 50% ACN/H$_2$O and oxidised in 0.1 M NH$_4$HCO$_3$ (1 mg/10 mL peptide concentration).

This protocol was used without modifications for the synthesis of **dOTK$^8$**, affording the product with 26% of yield (32 mg) and >95% purity (m/z: 1007.7 [M + H]$^+$).

**SPPS modifications for the synthesis of OTK$^8$[Aoa].** The first two amino acids, Fmoc-Gly-OH and Fmoc-Lys(Mtt)-OH were coupled to the resin, followed by deprotection of the Mtt group with 1% TFA in DCM and coupling of the Bis-Boc-aminooxy acetic acid to the ε-amino group of Lys$^8$. This was followed by treatment with DCM/Ac$_2$O/Et$_3$N (8:1:1, r.t.) for 30 min to cap any unreacted amino groups. SPPS was then continued following the standard protocol, affording the product with 15% of yield after HPLC purification (21 mg), and >95% purity by LC-MS (m/z: 1097.5 [M + H]$^+$).

**Synthesis of dOTK$^8$[SFB].** A solution of 50 mM **dOTK$^8$**, 200 mM DIPEA, and 400 mM SFB in DMF was prepared and incubated for 1 h (r.t.). The reaction was stopped by diluting the mixture with 0.05% TFA in H$_2$O (Solvent A) to a final concentration of 1% DMF and followed by RP-HPLC purification on an Agilent Eclipse XDB-C$_{18}$, 21.2 × 250 mm, 300 Å, 7 μm column with a 0.5%/min gradient of 0–50% Solvent B and a flow-rate of 15 mL/min (Solvent A: 0.05% TFA in H$_2$O; Solvent B: 90% ACN, 0.043% TFA), giving a > 95% purity (m/z: 1129.4 [M + H]$^+$).

**Synthesis of dOTK$^8$[ESF] and dOTK$^8$[ESF$_2$].** ESF was introduced to **dOTK$^8$** through a Michael addition at the ε-amino group of Lys$^8$. ESF attaches twice to the ε-amino group, hence equimolar amounts of **dOTK$^8$** (1 mM) and ESF (1 mM) in DMF for 1 h at r.t. were used to prepare **dOTK$^8$[ESF]**. **dOTK$^8$[ESF$_2$]** was instead obtained by using an 8-fold excess of ESF at r.t. overnight. Reactions were stopped by diluting the reaction mixture with H$_2$O and purified via HPLC using an Agilent Eclipse XDB-C$_{18}$, 21.2 × 250 mm, 300 Å, 7 μm column eluted with a gradient of 0-50% B, 0.5%/min at 15 mL/min (Solvent A: H$_2$O; Solvent B:

90% ACN, 10% H$_2$O), giving a > 95% purity (m/z: 1116.7 and 1227.8 [M + H]$^+$ respectively).

**Synthesis of OTK$^8$[Aoa-FDG].** A solution of **OTK$^8$[Aoa]** (1 mM) and **FDG** (5 mM) was prepared in buffer (100 mM sodium citrate, pH 4, 100 mM aniline) and incubated for 12 h (r.t.). This mixture was then diluted with 0.1 M NH$_4$HCO$_3$ (1:10, pH adjusted to 9) and left for oxidation at r.t. overnight. The peptide product was purified via HPLC using an Agilent Eclipse XDB-C$_{18}$, 21.2 × 250 mm, 300 Å, 7 μm column eluted with a gradient of 0–50% B, 0.5%/min at 15 mL/min (Solvent A: 0.05% TFA in H$_2$O ; Solvent B: 90% ACN, 0.043% TFA), giving a > 95% purity (m/z: 1259.4 [M + H]$^+$).

**Synthesis of OTK$^8$[Aoa-FBA].** Equimolar amounts of **OTK$^8$[Aoa]** and **FBA** were dissolved in DMF and incubated for 10 min (r.t.). The reaction was stopped by diluting the reaction mixture with 0.1 M NH$_4$HCO$_3$ (1:10, pH adjusted to 9) and left for oxidation at r.t. overnight. The peptide product was purified via HPLC using an Agilent Eclipse XDB-C$_{18}$, 21.2 × 250 mm, 300 Å, 7 μm column eluted with a gradient of 0–50% B, 0.5%/min at 15 mL/min (Solvent A: 0.05% TFA in H$_2$O; Solvent B: 90% ACN, 0.043% TFA), giving a > 95% purity (m/z: 1201.4 [M + H]$^+$).

**Stability test at different pH.** Tested peptides were dissolved (1 mM) in 100 mM sodium citrate buffer (pH 3), 100 mM potassium phosphate buffer (pH 7), or 100 mM sodium carbonate buffer (pH 9). 15 μL of each peptide solution was analysed by analytical C$_{18}$ RP-HPLC every hour for 10 h, recording the absorbance at 214 nm. HPLC conditions used: Vydac 210TP C$_{18}$ 5 μm, 2.1 × 250 mm column eluted with a gradient of 0–50% B, 1%/min, 0.25 mL/min (Solvent A: 0.05% TFA in H$_2$O; Solvent B: 90% ACN, 0.043% TFA).

**Serum stability.** One hundred and fifty microliters of rat or human serum (Sigma Aldrich) was preheated at 37 °C for 30 min. 25 μL of the tested peptide sample (0.3 M in 0.1 M potassium phosphate buffer at pH 7.2) or 0.1 M potassium phosphate buffer pH 7.2 (for blank sample) were added to the serum and incubated at 37 °C. 10 μL aliquots were taken at 0, 1, 2, 3, 4, 6, 12, 24 and 48 h and quenched with 35 μL ice-cold extraction buffer (50% ACN, 0.1 M NaCl, 1% TFA). Sampled were centrifuged at 20,000 × g for 10 min, and the supernatant analysed via LC-MS (column: Agilent Technology, Zorbex C$_{18}$, 2.1 × 100 mm, 5 μm, 300 Å. Elution method: 0–50% B in 50 min, 0.25 mL/min. Solvent A: 0.1% TFA in H$_2$O. Solvent B: 0.1% TFA in ACN). Each peptide sample was analysed in three independent experiments, and data analysis was performed using Prism Version 7, a nonlinear fit one-phase decay model.

**FLIPR Ca mobilisation assay.** COS-1 cells were cultured in Dulbecco's modified Eagle's medium (DMEM) and 5% foetal bovine serum (FBS). The cells were transfected 48 h before the assay with plasmid DNA encoding the hOTR, hV$_{1a}$R, or hV$_{1b}$R using FUGENE HD (Roche). A 1:6 ratio DNA (18 μg): FUGENE HD (108 μg) in 900 μL DMEM (serum-free) was added to the T75 flask of cultured cells. The cells were plated 24 h before the experiment at a density of 18,000 cells per well on black-walled 384-well imaging plates (Corning). On the day of the assay, the cells were washed with physiological salt solution (PSS; 140 mM NaCl, 11.5 mM glucose, 5.9 mM KCl, 1.4 mM MgCl$_2$, 1.2 mM NaH$_2$PO$_4$, 5 mM NaHCO$_3$, 1.8 mM CaCl$_2$, and 10 mM Hepes) for 5 to 10 min. Fluo-4-AM [4-(6-acetoxymethoxy-2,7-difluoro-3-oxo-9-xanthenyl)-4′-methyl-2,2′-(ethyle-nedioxy)dianilineN,N,N′,N′-tetraacetic acid tetrakis(acetoxymethyl) ester] (4.8 mM) in PSS containing 0.3% fatty acid-free BSA was then added, and the cells incubated for 30 min at 37 °C. The cells were then transferred to a FLIPRTETRA (Molecular Devices) fluorescent plate reader to measure the Ca$^{2+}$ responses using a cooled charge-coupled device (CCD) camera with excitation at 470 to 495 nm and emission at 515 to 575 nm. The baseline fluorescence was set to a minimum of 1000 arbitrary fluorescence units. The baseline fluorescence was read 10 times, followed by the addition of

test compounds and fluorescence measurement every second for 180 s. The test compounds were added as an 8-point gradient in triplicate, with the highest concentration of 10 μM in PSS. To convert the raw fluorescence data to DF/F, the baseline fluorescence readings were subtracted from subsequent time points, and the resulting value was divided by the baseline fluorescence values. For the concentration-response curves, the maximum DF/F values were plotted against the test compounds concentration and normalised to the response elicited by the native ligand (OT for hOTR and vasopressin for $hV_{1a}R$ and $hV_{1b}R$). The potency ($EC_{50}$) for each test compound was determined by nonlinear regression (four-parameter Hill equation with a Hill coefficient of 1, fitted to the data using GraphPad Prism version 4.00).

### Radiochemistry and imaging

**Materials and methods.** **OT** and **dOTK$^8$** were synthesised using Fmoc-SPPS. [$^{18}$F]**SFB** was produced with minor modifications from literature procedures[56] on a Synthra GP Extent system (*SI-2*). All solvents were purchased from Merck and were of reagent or HPLC grade. $C_{18}$ SepPak cartridges were purchased from Waters. Microfluidic radiochemistry was performed on an Advion NanoTek system, using the already published radiopharmaceutical optimisation[57] and production[43] methods. A Shimadzu chromatographic system, comprising of a CBM-20 controller, LC-20AD pump, SIL-20AHT auto injector SPD-M20A PDA, LabLogic Posi-RAM gamma detector, and RP column was used for the HPLC analysis. Analytical HPLC elution methods were:

- Method A: 2 mL/min; 2 min 95% [$H_2O$ with 0.05% TFA]/5% [ACN with 0.05% TFA]; 15-minute ramp to 5%/95%; 0.5 min recondition to 95%/5%. Phenomenex Luna 5 μm C18(2), 100 Å, 150 × 4.60 mm, 5 μm column.
- Method B: 2 mL/min; 2 min 95% [$H_2O$ with 0.05% TFA]/5% [ACN with 0.05% TFA]; 15-min ramp to 5%/95%; 0.5 min recondition to 95%/5%. Chromolith High Resolution RP-18e 50 × 4.6 mm column.
- Method C: 1 mL/min; 80% [$H_2O$ with 0.05% TFA] / 20% [ACN with 0.05% TFA]. Chromolith High Resolution RP-18e 50 × 4.6 mm column.

For the biodistribution and imaging studies, all animal procedures were approved by the Animal Ethics Committee of the University of Sydney and conducted in accordance with protocol number 2017/1194.

**Vessel conjugation testing.** 3.5–4.5 GBq of non-carrier added (nca) [$^{18}$F]**SFB** was obtained from the Synthra system in 1 mL of ACN, while 1 mg of **dOTK$^8$** or **OT** was dissolved into 1 mL of 0.05 M $Na_2HPO_4$ (pH 8.5). An aliquot of 50 μL for each solution was added into the insert of an HPLC vial that was placed into a heating block set at 25, 50, or 70 °C. The reactions were quenched at 10 or 20 min by adding 20 μL of ACN, and 5 μL of this mixture was analysed by HPLC, Method A.

**Microfluidic conjugation testing.** 3.5–5 GBq of nca [$^{18}$F]**SFB** obtained from the Synthra system in 1 mL of ACN, were loaded into the storage loop of an Advion system (P3). One milligram of **dOTK$^8$** or **OT** was dissolved in 1 mL of 0.05 M $Na_2HPO_4$ (pH 8.5), and such solution was loaded in the storage loop of P1. Reactions were performed by delivering 5–50 μL of both reactants into PEEK coiled-tubing microfluidic reactor (I.D. 250 μm, 1.15 m length, 56 μL volume) kept at constant temperature. The system allowed the testing of different reaction conditions by performing several reactions in a sequential manner, varying at each iteration the following parameter: flow rate (for each reagent, 5–50 μL/min), reactor temperature (25–90 °C), reagent solutions ratio (0.25–4). Reactant solutions were pushed through the fluidics using $H_2O$ as the inert solvent, and each reaction mixture was collected separately at the outlet of the reactor and analysed by Radio-HPLC (Method A or B).

**Microfluidic production of [$^{18}$F]dOTK$^8$[SFB].** [$^{18}$F]**SFB** was prepared following standard procedures and obtained in 1.5 mL of ACN in a

quantity of 19–23 GBq; this represented the radiolabelling solution that was loaded in the storage loop (~500 μL) of P3. 1 mg of **dOTK$^8$** was dissolved in 1 mL of 0.05 M $Na_2HPO_4$ (pH 8.5) as stock solution; 0.5 mL of this stock was diluted with additional 0.25 mL of 0.05 M $Na_2HPO_4$ representing the precursor solution and was loaded into the storage loop of P1 (~500 μL). Before actual production, two 'dummy' reactions were performed, using 30 and 10 μL each, and used to estimate radiochemical conversion by HPLC. For production, 150–200 μL of both radiolabelling and precursor solutions were delivered into the two inlets of the PEEK microfluidic coiled-tube reactor heated at 50 °C at a cumulated flow rate of 20 μL/min. The reaction mixture was collected in the loop of an HPLC injector (~500 μL). The content of such loop was injected in a Chromolith RP semi-prep column (150 × 10 mm, 13 nm pores), using a 25/75 = ACN/$H_2O$ with 0.05% TFA (v/v) at a 2.5 mL/min flow. The [$^{18}$F]**dOTK$^8$[SFB]** peak, eluting at times ranging 7–9 min, was detected using a Carroll-Ramsey 105S-1 high-sensitivity probe system and collected by routing the product fraction through a pre-conditioned Waters SepPak $C_{18}$ Plus cartridge. The cartridge was rinsed with 1 mL of $H_2O$ and dried with $N_2$ flow; the product was then eluted with 750 μL of EtOH in a COC plastic V-vial. The solvent was then evaporated under a stream of $N_2$ and heating at 60 °C in 15 min; the final residue, typically ~50 μL of $H_2O$, was recovered into a single-use 1.5 mL Eppendorf tube by dilution with isotonic saline to the volume required for imaging. This full production process could be repeated sequentially one more time, thus providing a second batch of product from the same starting materials; using this process, the 1st and 2nd batches were obtained after 1.5 and 2.5 h, respectively, from [$^{18}$F]**SFB** EOS. After productions, a further 'dummy' run of 10 μL could be performed, in order to confirm the estimated RCY from the collected radioactivity.

Without taking into account the distinction across 1st and 2nd batches, the activity produced was 148 ± 75 MBq for each 100 μL of [$^{18}$F]**SFB** used (25 ± 7% RCY), with a $A_m$ = 191 ± 97 MBq/nmol (range: 80–460) and radiochemical purity >95% (n = 13). The final product was analysed with HPLC using Method B for assessing radiochemical and chemical purity and Method C to determine $A_M$. Residual solvents were analysed by GC-MS, and the residual amount of EtOH and ACN were, on average, 0.32 and 0.05, respectively (% w/v).

**Vessel production of [$^{18}$F]dOTK$^8$[SFB].** Reagents were prepared for the FlexLab synthesiser (iPhase) and placed in vials and positions as follows (*SI-5*). A solution of 1 mg of **dOTK$^8$** precursor was dissolved in 500 μL of 0.05 M potassium phosphate buffer (pH 8.5) and added to Vial 5, as close as possible to [$^{18}$F]**SFB** transfer. Vial 2 was loaded with 500 μL of ACN. Vial 4 was loaded with 1 mL of ultra-pure $H_2O$. A Waters $C_{18}$ Light SPE cartridge was preconditioned with 5 mL EtOH followed by 10 mL isotonic saline, and placed in position SPE-D. Vial 11 was loaded with 4.75 mL isotonic saline. Vial 12 was loaded with 250 μL of EtOH, and vial 13 was loaded with 5 mL of ultra-pure $H_2O$. HPLC Collection Vessel 2 was loaded with 25 mL of ultra-pure $H_2O$.

[$^{18}$F]**SFB** solution in ACN was transferred to FlexLab Reactor 1 via He push and dried under vacuum for 6 min at 80 °C. The reactor was cooled to 25 °C before the addition of the 500 μL precursor solution followed by 500 μL of ACN. The reaction mixture was magnetically stirred and heated to 40 °C for 15 min before being cooled to 25 °C. 1 mL of ultra-pure $H_2O$ was added to dilute the reaction, and the mixture was injected onto the loop of a semi-prep HPLC. The purification was conducted on a Sunfire $C_{18}$ column (250 × 10 mm) using isocratic conditions (70% $H_2O$, 25% ACN, 5% MeOH, 4 mL/min). The product fraction eluted in the 22–24 min range and was collected into HPLC Collection Vessel 2. The diluted fraction was concentrated on the $C_{18}$ Light SPE cartridge which was subsequently washed with 5 mL of $H_2O$. The product was eluted from the SPE into the final product vial with 250 μL EtOH and subsequently 4.75 mL isotonic saline, yielding the final product in 5 mL (5% EtOH in saline). Starting from an average of 35.9 ± 5.1 GBq of [$^{18}$F]**SFB**, we obtained 4.4 ± 0.5 GBq of [$^{18}$F]

**dOTK$^8$[SFB]** 60 min after [$^{18}$F]SFB EOS (RCY of 17.8%), with >95% radiochemical purity and A$_m$ of 364 ± 65 MBq/nmol ($n = 8$).

**Radiotracer biodistribution and pharmacokinetic study.** The pharmacokinetic and biodistribution studies were performed in separate cohorts. For the PK studies, 5 rats were injected *via* lateral tail vein with 30–60 MBq (0.2–0.6 nmol) of [$^{18}$F]**dOTK$^8$[SFB]**, and the arterial blood samples were taken at regular intervals over 1 h post-injection. The animals were euthanised 1 h after injection, and tissue samples were harvested and wet-weighed. The collected blood, plasma, and tissues collected were counted in an automatic gamma counter (Wizard 2, Perkin Elmer). The results of the PK study were recorded as %ID/g for both blood samples and the organs (mean ± SEM).

**Micro-PET/CT imaging.** Twenty two male SD rats were used for the baseline, competition, and blocking study. For the PET/CT scans, rat were first anaesthetised with isoflurane (5% induction, 1.5–2% maintenance) and positioned in the gantry of the Siemens Inveon PET/CT Scanner (Siemens), followed by lateral tail vein injection of [$^{18}$F] **dOTK$^8$[SFB]**; a 60 min dynamic PET scan was started 10 s prior to the injection (40–60 MBq, 0.3 mL). Except for pilot experiments, the radiotracer injection was spiked with non-radioactive isotopologue to achieve a total injected mass of 1 nmol. The injection was performed by using a 26 G needle fitted into a silicon cannula (0.3 mm I.D.) connected to a microsyringe pump. The needle was placed 2–3 mm into the vein and held in place with Vetbond. Thirty microliters of saline were used to flush the remaining activity in the cannula. The rats in the blocking group received 100 μg of non-radioactive isotope analogue or OT in 100 μL saline, injected *via* the previously installed cannula, 30 min before the PET scan (20 min prior in the pilot experiment). The rats in the competition experiments received the same dose used for blocking, but 60 min post radiotracer injection, and were scanned for additional 30 min. A short CT scan was performed at the end of all PET imaging sessions to acquire anatomic details and for attenuation correction. Image reconstruction was performed using IAW 2.02 software (Siemens). The list mode data was histogrammed into 27 dynamic frames or 48 dynamic frames for displacement scans. Emission sinograms were reconstructed using 2D-filter back projection with a zoom of 1. The reconstructed images consisted of a 256 × 256 × 159 matrix, with a voxel size of 0.388 × 0.388 × 0.80 mm$^3$. The data were corrected for attenuation, scatter, randoms, normalisation, isotope decay, branching ratio, and dead time and were calibrated to Bq/mL. PET and CT images were fused using the Inveon Research Workplace software 4.2 (Siemens), and 3D ROIs were manually created on organs of interest, while the accumulation of $^{18}$F was quantified as % injected dose per gram (%ID/g). Time activity curves (TACs) were plotted using MS Excel (Version 2310, Microsoft Corporation, USA).

**Patlak plot.** The data handling and Patlak algorithm implementation were performed using in-house developed MATLAB scripts (version 9.13.0, R2022b). The $K_i^{ref}$ parameter was estimated using a weighted least square approach, where the elements of the diagonal weight matrix are expressed by $\frac{C_{ROI}(t_i)}{\Delta(t_i)}$, $i \in [t^{\cdot}, t^{\cdot} + n]$ and $\Delta(t_i) = t_i - t_{i-1}$. The statistical analysis was conducted using the mean of the $K_i^{ref}$ estimated using the last $n = 4$, 5, and 6 samples of the Patlak plot.

### Reporting summary
Further information on research design is available in the Nature Portfolio Reporting Summary linked to this article.

### Data availability
All data associated with this study are present in the paper or the Supplementary Information and Video. Samples of compounds reported are available upon request, depending on available supply and under condition of a signed material transfer agreement.

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

## Acknowledgements

The work has been funded by the ARC Linkage project LP150101307. MM was supported by the European Research Council under the European Union's Horizon 2020 research and innovation programme (714366) and by the Australian Research Council (FT210100266). The authors acknowledge the facilities and scientific and technical assistance of the National Imaging Facility, a National Collaborative Research Infrastructure Strategy (NCRIS)

capability, at the National Research Cyclotron in Camperdown, NSW, operated by ANSTO. The contribution of Rajeev Sheth, Jarrod Pynt, Michael Tran, and Tien Pham is kindly acknowledged. In addition, we would like to thank Gary Perkins and Nikolas Paneras for helping in the initial setup of the project and Kelly Smart for the useful discussions.

## Author contributions

Conceptualisation: G.P., R.B., M.C.G., I.H. and A.G. Methodology: G.P. and M.M. (chemistry), G.P. (radiochemistry), G.P., R.B., M.C.G., A.P., L.Y. and A.G. (imaging). Investigation: M.M. and A.H. (peptide synthesis), N.T. (pharmacology tests), G.P., B.Z., T.M., L.M., J.M. and M.K. (microfluidic production), I.G., L.S., A.M. and A.B. (vessel production), A.P., A.A., G.R., D.Z. H.H. (imaging experiments), A.P., A.A., D.Z. and S.Z. (data analysis). Supervision: M.M., G.P. and A.P. Writing—original draft: G.P., A.P., A.A., M.M., S.Z. and A.G. Writing—review and editing: All authors, except L.Y.

## Competing interests

The authors declare no competing interests.
