## [Transparent Peer Review file · Communications Chemistry]

Development and characterization of novel oxytocin analogues for PET imaging

Corresponding Author: Professor Giancarlo Pascali

Version 0:

Reviewer comments:

Reviewer #1

(Remarks to the Author)

The work described in the paper is very preliminary and premature. Important data are missing (binding affinity, in vivo binding to OXTR etc.) to confirm binding to the OXTR, the central assumption of the paper.

SPECIFIC COMMENTS:

- Consensus nomenclature for radiopharmaceuticals should be followed
- Line 73: 'in several animal studies', but only 1 reference.
- Fig 1A should display the oxime formation before disulfide bond formation
- OTK8[Aoa-FBA] and OTK8[Aoa-FDG] were not further pursuit for radiolabeling with 18F.
- Stability in buffered solutions at different pHs does not translate do stability in biological fluids (Line 143/144)
- Stability assessment: remove 'tracer' everywhere where non-radioactive compounds are used
- SUPpl Fig 1 change intact tracer to intact compound
- Metabolite is SI-3 and not SI-1

- Metabolite analysis: were metabolites identified by coinjection with the proposed Metabolite reference?

- Line 162: define abbreviation VP
- Primary data for FLIPR calcium mobilization data missing

- No binding data

- Conventional and microfluidic radiolabeling approach: n=1 not sufficient, reproducibility questionable

- RCY are RCCs for 18F-SFB-incorporation and not isolated RCYs

- Add yields, total synthesis time and QC for 18F-SFB syntheses into SI

- Larger bolus at the same prec concentration?

- Rephrase sentence line 241-243, clarifying which reaction mixture contains which compounds

- Why is there [18F]FBA in the [18F]SFB reaction with dOTK8? Please clarify in text and Figure 4. Add HPLC traces of non-radioactive reference compounds

- Fig 3 and 4: time axis is missing

- Only 2 full production processes from the same batch of 18F-SFB.

- Reproducibility questionable for all optimization radiolabeling strategies, as well as final production stage (n=1, n=2 for final production)

- RCY calculation inaccurate via Synthra-detectors due to 'shine' of radioactivity in the system

- SI: 4. HPLC profiles, chromatograms with multiple peaks should have labels on each peak with the identity of the compound
- Line 303-305: what is the purpose of using a microfluidic approach, if the standard vessel approach in a synthesizer provides similar yields and purity.
- Total synthesis time, HPLC-QC (RCP) and GC results are not reported in the 'Full production and QC' section
- Discussion around radiolabeling approaches missing, References incomplete
- %ID/g uptake in healthy rats seem very low, no brain uptake
- Fig 6: Y-axes definition is missing
- Conclusion: Line 402: five tracers were not tested, only 1 tracer was tested; line 410: process length was not mentioned in the Results and Discussion section
- A disease preclinical model should be the next step ahead of clinical translation.
- 18F-SFB-labeled OT is produced with conventional method. Why is this tracer not evaluated in comparison to 18F-dOTK8[SFB]?
- Spelling mistakes

Experimental:

- Amounts and yields for synthesized compounds are missing, Were peptides used as crude or HPLC-purified (dOTK8, OTK8[Aoa]?)
- Retention time HPLC for non-radioactive material
- Ca-mobilization assay does not determine receptor affinity
- 18F-SFB-labeling of OT described in experimental
- 18F-dOTK8[SFB] 6-8min retention time on HPLC described in Experimental, does not correlate with semi-prep. HPLC chromatogram in Fig 4a.

Reviewer #2

(Remarks to the Author)

The manuscript by Giancarlo Pascali et al. presents a comprehensive study on the development and characterization of oxytocin analogs for PET imaging. The authors designed and synthesized five derivatives, evaluating their stability and other key properties to identify the most promising candidate for radiosynthesis and in vivo assessment. Their findings demonstrate radiotracer accumulation in the pituitary gland and mammary pads, highlighting the potential of these novel oxytocin derivatives for imaging applications. This study represents a well-rounded and thorough approach to the development of oxytocin-based PET tracers and makes a valuable contribution to the field. I recommend acceptance after revision, provided the following comments are addressed:

Comments:

1. In the synthesis and design of oxytocin derivatives, it is unclear how the amine groups from cysteines were removed. The manuscript mentions the use of dCys, which is somewhat confusing, as the term typically refers to the D-isomer of cysteine. Please clarify whether the amine group was specifically removed or if the entire amino acid was replaced or modified.
2. The authors have used HPLC to determine the hydrophobicity of oxytocin derivatives. However, hydrophobicity is typically assessed using the water-octanol partitioning system. It would be more informative to include cLogP determination using the water-octanol system for a more conventional and standardized evaluation of hydrophobicity.
3. The manuscript begins by discussing intranasal administration but ultimately focuses on intravenous (IV) administration. Have the authors attempted intranasal administration of the radiotracer? If so, how does its pharmacokinetics compare to intravenous administration? Providing data or discussion on this aspect would enhance the relevance of the study to intranasal delivery applications.

Reviewer #3

(Remarks to the Author)

Review and comments to the authors

The authors present the synthesis and pharmacological characterization of five oxytocin-like peptides and selected the most suitable candidate for development as fluorine-18 labeled radiotracer for preclinical investigation of the oxytocin/oxytocin receptor signaling system by PET imaging.

In my opinion the manuscript is very well written, the content is within the scope of the journal and will be of interest to scientist working in the field of PET radiotracer development. The experiments are well designed and the presented results

are supported by the data obtained. Therefore, I propose to publish the article after some minor corrections.

General remark:

There are only a few references in the manuscript directing the reader to the Supplementary Information and most of them are incorrect (e.g. line 204, line 217, line 428). As the content of the SI is important for the manuscript, I would like to recommend to refer to each section and each figure presented in the SI at an appropriate position in the manuscript.

Detailed remarks:

- line 117: The five peptides shown in Figure 1A are not yet tracers, same in line 167.
- line 162: Please add vasopressin for VP.
- line 165-166: Please briefly explain the principle of the assay.
- line 167: I think you cannot state here, that a potency of 5-47 nM is sufficient for PET imaging, because that is what was actually also investigated in this study.
- line 193: The authors mention the "direct labeling of OT". Please include and name this radiotracer in Figure 1 or as a new scheme in the SI.
- Figure 3 A and B: It is not the RCY of OT or dOTK8 shown but of the corresponding 18F-labeled tracers. Please correct the axis labels.
- lines 251-253: I suggest adding a sentence describing the observation of differences in the elution order of F2 and F3 in the semi-prep and analytical chromatograms.
- Figure 4: Please include the conditions for semi-prep HPLC and analytical HPLC in the Figure caption.
- lines 267-269: I recommend to include here information on the sorption and recovery to demonstrate that there is no considerable loss of activity due to the simplified process.
- line 302: According to the nomenclature rules published in 2017 (<https://doi.org/10.1016/j.nucmedbio.2017.09.004>), I recommend to use A_m instead of A_M as abbreviation for molar activity. Please change it accordingly throughout the manuscript and SI.
- SI, Section 3, "Bioconjugation experiments using [18F]ESF". I think you mean [18F]SFB? Please change in the whole chapter.
- SI, Section 6, "Calibration curve for [18F]dOTK8[SFB]". This is not a calibration curve for the radiotracer but for the non-radioactive compound. Please correct accordingly.
- SI, Chromatograms SupplInfo 8-11: Please indicate the desired products in the chromatograms.

Version 1:

Reviewer comments:

Reviewer #2

(Remarks to the Author)

The authors have satisfactorily addressed all of my comments. I recommend acceptance of the manuscript in its current form.

Also, I believe the concerns raised by Reviewer 1 have been satisfactorily addressed by the authors. Based on their responses, I have no further comments and recommend the acceptance of this manuscript in its current form.

Reviewer #3

(Remarks to the Author)

In my opinion, the manuscript can be published now.

We thank the reviewers for their time, effort, and thoughtful comments on our manuscript.

Reviewer #1 (Remarks to the Author):

The work described in the paper is very preliminary and premature. Important data are missing (binding affinity, in vivo binding to OXTR etc.) to confirm binding to the OXTR, the central assumption of the paper.

We thank the reviewer for their comment. The primary focus of this work was the development of an OT-like PET tracer, encompassing its rational design, synthesis, radiolabelling, production, and comprehensive biophysical and pharmacological characterisation. As such, we believe the tracer development pipeline presented here is mature and appropriate for publication in Communications Chemistry and highlights innovative molecular tools for broader applications. While translational imaging studies, including human PET, are underway, they are outside the chemistry focus of this manuscript and the journal.

In line with our design goal to create an OT-like tracer, we prioritised functional assays over binding assays to provide more relevant pharmacological information, including bioactivity and information on partial/full agonism. The low nanomolar EC₅₀ values, OT-like tracer design, and side-by-side comparison with endogenous OT demonstrate direct receptor engagement and OTR activation. *In vivo* PET imaging furthermore confirmed uptake in OTR-rich tissues, and competition studies with OT and the non-radioactive isotopologue underpin receptor-specific accumulation.

Given these converging lines of results and structural similarity to OT, we believe additional radioligand displacement assays are not necessary and would not change the outcome or interpretation of our designs and data. We respectfully ask that this rationale be taken into consideration.

SPECIFIC COMMENTS:

- Consensus nomenclature for radiopharmaceuticals should be followed

We have revised the manuscript to follow standard radiopharmaceutical nomenclature. For example, we changed “A_M” into “A_m”, and used “fluorine-18”, “radiofluorination” or similar text instead of “¹⁸F” when such word is used as an adjective.

- Line 73: ‘in several animal studies’, but only 1 reference.

We extended the example list, including some recent reviews on this topic, and highlighted the non-human primate studies. We have modified the text as follows:

“OT brain penetration has been investigated in several animal studies,^{9,14} including non-human primates,^{15”}

- Fig 1A should display the oxime formation before disulfide bond formation

We have modified the figure accordingly.

- OTK8[Aoa-FBA] and OTK8[Aoa-FDG] were not further pursued for radiolabeling with ^{18}F .

Correct. These analogues were not pursued due to the observed instability of the aminoxy group during oxidative folding and the requirement for a faster labelling strategy compatible with the short half-life of ^{18}F . This rationale is now further clarified in lines 131–133, visualised in Fig. 2, and discussed in the “Justified selection of lead OT analogue for radiolabelling” section.

“OTK⁸[Aoa-FDG] was the second-best candidate; however, like OTK⁸[Aoa-FBA], it raised concerns due to the need for optimisation of radiolabelling via oxime ligation and its limited serum stability.”

- Stability in buffered solutions at different pHs does not translate to stability in biological fluids (Line 143/144)

We modified this sentence and removed biological fluids.

“First, candidate tracers were exposed to buffered solutions at pH 3, pH 7 and pH 9 to assess their pH stability, using analytical C18-RP-HPLC over a period of 10 h.”

- Stability assessment: remove 'tracer' everywhere where non-radioactive compounds are used

“Tracer” has been removed and replaced with more accurate terminology for non-radioactive compounds where appropriate.

- Suppl Fig 1 change intact tracer to intact compound

Fixed.

- Metabolite is SI-3 and not SI-1

Fixed.

- Metabolite analysis: were metabolites identified by coinjection with the proposed Metabolite reference?

Metabolites were identified by LC-MS based on peptide sequence, observed mass, and reasonable enzymatic cleavage patterns, but not confirmed by co-injection with a synthetic version of this metabolite. We added further information in the relevant section, clarifying this.

“The main metabolite identified by LC-MS analysis was the dCys/Cys-Tyr-Ile-Gln-Asn-Cys-Pro macrocycle (SI-3), based on matching mass and reasonable C-terminal enzymatic cleavage.”

- Line 162: define abbreviation VP

Vasopressin (VP) was added.

- Primary data for FLIPR calcium mobilization data is missing.

The person who conducted the FLIPR experiments left the lab five years ago, and we no longer have access to her primary data set; we only possess the previously calculated EC₅₀ and SEM values in table form presented in Fig. 2C.

- No binding data.

We prioritised functional assays over binding studies to provide data on OT-like receptor engagement, activation, and partial/full agonism. The low nanomolar EC₅₀ values along with the OT-like structure support direct receptor engagement and activation at the tested OTR and VPRs.

- Conventional and microfluidic radiolabeling approach: n=1 not sufficient, reproducibility questionable

In order to achieve more efficient optimisation experiments with reduced operator exposure, we chose to explore a broader range of reaction conditions, rather than to replicate a narrow set of optimisation conditions. This approach is based on the bias/variance trade-off, whereas a larger parameter set allows for a more accurate optimisation model, even with a lower precision for each experimental point. Importantly, the reproducibility of the optimised conditions was demonstrated in the successive production runs, with N=9 for the first and N=4 for the second batch, as detailed in the “Full production process and quality control” section. These repeated syntheses yielded consistent radiochemical yields and molar activities, confirming the robustness and reliability of the radiolabelling approach.

- RCY are RCCs for 18F-SFB-incorporation and not isolated RCYs

While we agree that the reported values are not "isolated RCYs", the reported values refer to the radiochemical yields (RCYs) for the specific process under study, i.e. the sole incorporation of [¹⁸F]SFB in the product, as assayed by radio-HPLC analysis of the reaction mixture. To add on this, given the use of PEEK microreactor (vs silica), we also verified that >95% of radioactivity was recovered from the system for each reaction, thus justifying the adoption of the RCY term. To address this point more clearly, we modified the text where RCY is first introduced to provide further context.

“Radiochemical yields (RCY)³⁶ of these reactions were determined by radio-HPLC, based on the initial [¹⁸F]SFB activity and calculating the peak area of [¹⁸F]dOTK⁸[SFB] relative to the areas of all other radioactive peaks (Fig. 3A).”

- Add yields, total synthesis time and QC for 18F-SFB syntheses into SI

We added this information in the relevant SI section.

“The activity yield of [¹⁸F]SFB was 24±3% for a 60-minute synthesis process, determined using Synthra radiodetectors. Radiochemical purity was confirmed to be >95% in initial validation runs by radio-HPLC (Column: Phenomenex, Luna 5μ C₁₈ 100Å, 150 x 4.6 mm, 5 μm; Mobile Phase: Acetic acid:water:acetonitrile (0.1:65:35); Flow rate: 1 mL/min). [¹⁸F]SFB was used directly in the subsequent [¹⁸F]dOTK⁸[SFB] production process without further quality controls..”

- Larger bolus at the same prec concentration?

In the microfluidic setup, we used larger bolus volumes for both [¹⁸F]SFB and dOTK⁸ (100–200 μL at production scale) compared to the optimisation phase (20–40 μL). While the absolute amount of precursor (dOTK⁸) was consequently higher in the larger bolus, its concentration should be maintained at 1 mg/mL. For practical reasons (i.e. complete recovery of substrate solution during microreactor system loading), the actual value resulted to be 0.7 mg/mL in the production phase, as detailed in the experimental section. We hope this clarifies the rationale behind the use of larger bolus volumes while maintaining the precursor concentration in line with the optimisation phase.

“The following preferred reaction conditions were adopted: a solution of [¹⁸F]SFB (>15 Gbq in 1.5 mL of CH₃CN) and a solution of dOTK⁸ (0.5 mg in 0.75 mL of 0.05 M Na₂HPO₄, pH 8.5) were delivered at 10 μL/min each in the 56 μL PEEK microreactor kept at 50°C.”

- Rephrase sentence line 241-243, clarifying which reaction mixture contains which compounds

We rephrased the sentence to clarify, please see the reply just below.

- Why is there [18F]FBA in the [18F]SFB reaction with dOTK⁸? Please clarify in text and Figure 4. Add HPLC traces of non-radioactive reference compounds

Hydrolysis of [¹⁸F]SFB into [¹⁸F]FBA is a common side reaction. We now mention this in the text.

“Since the reaction mixture of the conjugation reaction comprised four main radioactive components due to incomplete conjugation and hydrolysis, namely [¹⁸F]SFB, [¹⁸F]dOTK⁸[SFB], [¹⁸F]FBA and an unknown species, a semi-preparative RP-HPLC purification step was required to afford the desired radiotracer in sufficient purity for PET imaging (> 95%).

- Fig 3 and 4: time axis is missing

We added the time axis to Figure 4C & D.

In Figure 3, the x-axis is temperature and flow rate, not time.

Only 2 full production processes from the same batch of ^{18}F -SFB.

We can produce two batches of ^{18}F]dOTK⁸[SFB] from a single batch of ^{18}F]SFB on the same equipment back-to-back. We performed such a back-to-back production 4 times. This approach is highly innovative due to its complete automation and reliability, in line with the “dose-on-demand” strategy reported by our group previously.

- Reproducibility questionable for all optimization radiolabeling strategies, as well as final production stage (n=1, n=2 for final production)

The conjugation reaction step performed at production levels of radioactivity gave a $46 \pm 8\%$ RCY, which aligned well with the 47% obtained during the optimisation phase. The full microfluidic production process, comprising also semi-prep HPLC purification and SPE formulation, produced 9 1st batches and 4 2nd batches, with an average $25 \pm 7\%$ RCY (from ^{18}F]SFB). This value is slightly higher than the same RCY obtained in a standard batch FlexLab synthesiser, 17.8%, and lower than the RCY of the sole conjugation reaction step, which is expected given the losses commonly experienced during purification and formulation.

- RCY calculation inaccurate via Synthra-detectors due to ‘shine’ of radioactivity in the system

We assessed the RCY of the final product using both i) the “dummy” method (as described in Ref 42), which is well suited for flow microfluidic systems, and ii) an estimate based on the starting ^{18}F]SFB activity as measured by the Synthra radiodetectors. The two RCY values (25% vs 19%) were in a comparable range. However, the lower RCY calculated using the Synthra detector may reflect an overestimation of the starting ^{18}F]SFB, consistent with the reviewer’s observation. We have now added this clarification to the manuscript.

“On average, 149 MBq for each 100 μL of ^{18}F]SFB was obtained, corresponding to 169 MBq for the 1st batch (N=9) and 104 MBq for the 2nd batch (N=4). RCY was assessed using two approaches. First, based on starting ^{18}F]SFB using the “dummy” method,⁴⁴ resulting in a value of $25 \pm 7\%$. Secondly, RCY was also calculated based on the starting amount and volume of ^{18}F]SFB, as measured by the Synthra radiodetector, which resulted in a lower RCY of $19 \pm 8\%$. This small discrepancy may be due to an overestimation of the starting ^{18}F]SFB caused by detector spillover signal (i.e. radioactive shine) in the Synthra system. Nevertheless, the agreement between these two independent assessment

methods supports the reliability of the reported RCY values and demonstrates that very little radioactivity was lost in the microfluidic system."

- SI: 4. HPLC profiles, chromatograms with multiple peaks should have labels on each peak with the identity of the compound

We revised the captions of all the chromatograms to indicate the radioproduct peaks assignment. We prefer this to improve the visibility of profiles, rather than including additional text elements in the graphs.

- Line 303-305: what is the purpose of using a microfluidic approach, if the standard vessel approach in a synthesizer provides similar yields and purity.

The microfluidic approach was chosen for its key advantages during the optimisation phase of the project. Specifically, it enabled the rapid screening of multiple reaction conditions using minimal reagent volumes and radioactivity, thereby accelerating method development and conserving valuable materials. Additionally, once optimal conditions were identified at the analytical scale, they were translated to the production scale with good reproducibility. Since not every lab has microfluid setups, we also implemented the final production process on a standard synthesiser platform to provide protocols for tracer production using conventional radiosynthesis setups.

We now clarify this in the revised version.

"To enable production in laboratories without access to a microfluidic system, the synthesis of [¹⁸F]dOTK⁸[SFB] was successfully adapted to a conventional vessel-based method (i.e., using the Flexlab synthesizer), yielding comparable radiochemical purity and yield."

- Total synthesis time, HPLC-QC (RCP) and GC results are not reported in the 'Full production and QC' section.

Due to our dose-on-demand setup, it is challenging to report the total synthesis time in a conventional manner. However, to clarify, starting from a batch of [¹⁸F]SFB, the first production run requires ~1.5 hours, primarily due to the 30-minute loading step needed to fill the microfluidic storage loops. Each subsequent production run takes ~1 hour, as also noted in line 285 of the original manuscript.

To better clarify this, we revised the text in the "Full production and QC" section:

"The full production process—including microfluidic radioconjugation, HPLC purification, and formulation—was performed back-to-back, yielding two separate product batches from the same starting radioactivity and peptide precursor. Following the delivery of [¹⁸F]SFB to the microfluidic system, 30 minutes were required to load the storage loops with reagents and prepare the synthesizer. Once initiated, the production run took 60 minutes to complete. This process can be started at any time, provided a sufficient volume of radioactivity is available."

Regarding HPLC-based radiochemical purity (RCP) and GC results, these are included in the experimental section along with other key production details. We feel this is the most appropriate location for such technical information, allowing the main manuscript to retain a clearer narrative flow.

- Discussion around radiolabeling approaches missing, References incomplete

We reported a brief discussion around radiolabelling methods in the Conclusions section and we have now added a key reference on radiolabelling approaches to peptides.

"This study focused on developing an OT-like radiotracer to enable PET imaging of the OT/OTR system, thereby expanding the molecular toolbox for investigating this ancient and multifaceted GPCR.² Following conventional peptide labelling strategies,⁵¹ we designed and tested five OT analogues, and finally selected [¹⁸F]dOTK⁸[SFB] as the best lead for PET imaging based on physicochemical, pharmacological, radiochemical, and practical considerations."

- %ID/g uptake in healthy rats seem very low, no brain uptake

We agree that the %ID/g values are low; however, this is not unexpected for a peptide tracer, especially when not targeting receptors overexpressed in disease conditions. Examples of low %ID/g values can be found in Evaluation of 68Ga-Radiolabeled Peptides for HER2 PET Imaging- PubMed or PET Imaging of FSHR Expression in Tumors with 68Ga-Labeled FSH1 Peptide- Pan- 2017- Contrast Media & Molecular Imaging- Wiley Online Library.

As for the brain uptake, the study's aim was not to develop a tracer capable of crossing the blood-brain barrier, but to develop a radiotracer highly similar to native OT, which will enable studies investigating the biodistribution of administered OT (for example, for human intranasal delivery studies to investigate the time-dependent pharmacokinetics and biodistribution after delivery). We primarily used rat imaging here to demonstrate that our tracer is suitable for PET imaging, establish the radiolabelling and tracer preparation, and generate some preliminary OTR engagement data. Absence of detectable brain uptake was not unexpected, given the ongoing debate about the brain penetrance of OT and rodent to human species differences (e.g. see reference 9).

- Fig 6: Y-axis definition is missing.

We added %ID/g and min to the axis.

- Conclusion: Line 402: five tracers were not tested, only 1 tracer was tested; line 410: process length was not mentioned in the Results and Discussion section

Line 402: We have changed the word “tracers” to “analogues” here.

“...we designed and tested five OT analogues, and finally selected [¹⁸F]dOTK⁸[SFB]...”

Line 410: The term “process length” refers specifically to the total time required for the flow-based radiolabelling procedure, which includes both the chemical reaction (residence time) and the physical delivery of reagents through the microfluidic system. Unlike vessel-based chemistry, flow systems require balancing two interdependent time factors: residence time, which affects reaction efficiency (longer time typically increases RCY), and overall process time, which impacts radioactive decay (shorter time preserves more activity). These two are both governed by flow rate, and an optimal rate must be traded off between them. We have now clarified this more explicitly in the revised manuscript to avoid confusion.

"In addition, we demonstrated the importance of identifying an optimal flow rate for microfluidic production that balances RCY with overall process duration. We prioritized flow rates that offered slightly lower RCY in exchange for shorter production times, thereby improving reliability of activity yields and facilitating better management of imaging experiments."

- A disease preclinical model should be the next step ahead of clinical translation.

The OT-like tracer developed in this manuscript was developed for human PET imaging studies assessing the pharmacokinetics and biodistribution of various administration routes of OT in healthy humans. No preclinical disease models are required for such studies.

- 18F-SFB-labeled OT is produced with conventional method. Why is this tracer not evaluated in comparison to 18F-dOTK8[SFB]?

Direct radiolabelling of OT at its N-terminus (the only free amine) was conducted as a synthetic control experiment to validate the radiolabelling process. Such modifications to the N-terminus of OT are known to reduce OTR binding and activity; hence, we did not pursue biological evaluation. By contrast, [¹⁸F]dOTK⁸[SFB] was specifically designed to retain receptor engagement. We now clarify this better in the text.

"Direct radiolabelling of OT at its N-terminus using the same [¹⁸F]SFB conditions was performed as a synthetic control. However, due to the known sensitivity of the N-terminus to modification, which reduces both receptor binding and activity,³⁶ this construct was not pursued further for biological evaluation."

- Spelling mistakes

We have thoroughly checked the manuscript for spelling mistakes

Experimental:

-Amounts and yields for synthesized compounds are missing. Were peptides used as crude or HPLC-purified (dOTK8, OTK8[Aoa])?

The overall yield for dOTK⁸ was 26% after folding and HPLC purification, and the reaction was performed at the 0.125 mmol scale, yielding 32 mg. The overall yield for reduced dOTK⁸[Aoa] was 15% after HPLC purification, yielding 21 mg. This info is now reported in the experimental section.

"This protocol was used without modifications for the synthesis of dOTK⁸, affording the product with 26% of yield (32 mg) and >95% purity (m/z: 1007.7 [M+H]⁺)."

"SPPS was then continued following the standard protocol, affording the product with 15% of yield after HPLC purification (21 mg), and >95% purity by LC-MS (m/z: 1097.5 [M+H]⁺)."

-Retention time HPLC for non-radioactive material

The retention times of the non-radioactive reference compounds were analysed and aligned with those of the corresponding radioactive products (within the expected detector's time gap). This alignment was used to support peak identification in the radio-HPLC traces. One radioactive by-product could not be matched to any anticipated non-radioactive by-product and remains unidentified.

-Ca-mobilization assay does not determine receptor affinity

We removed 'of receptor affinity' in the experimental subtitle (line 530). It now states: *"FLIPR Ca mobilisation assay"*

-18F-SFB-labeling of OT described in experimental

The labelling of OT is reported in the experimental sections "Vessel conjugation testing" and "Microfluidic conjugation testing".

"...while 1 mg of dOTK⁸ or OT was dissolved into 1 mL of 0.05 M Na₂HPO₄ (pH 8.5)"

"1 mg of dOTK⁸ or OT were dissolved in 1 mL of 0.05M Na₂HPO₄ (pH 8.5)..."

-18F-dOTK8[SFB] 6-8 min retention time on HPLC described in Experimental, does not correlate with semi-prep. HPLC chromatogram in Fig 4a.

We thank the reviewer for pointing out this inconsistency; the text has been modified to "...7-9 min..."

Reviewer #2 (Remarks to the Author):

The manuscript by Giancarlo Pascali et al. presents a comprehensive study on the development and characterization of oxytocin analogs for PET imaging. The authors designed and synthesized five derivatives, evaluating their stability and other key properties to identify the most promising candidate for radiosynthesis and in vivo assessment. Their findings demonstrate radiotracer accumulation in the pituitary gland and mammary pads, highlighting the potential of these novel oxytocin derivatives for imaging applications. This study represents a well-rounded and thorough approach to the development of oxytocin-based PET tracers and makes a valuable contribution to the field. I recommend acceptance after revision, provided the following comments are addressed:

We thank the reviewer for taking the time to go through the manuscript and for the useful feedback and comments, which we have addressed in the revised manuscript.

Comments:

1. In the synthesis and design of oxytocin derivatives, it is unclear how the amine groups from cysteines were removed. The manuscript mentions the use of dCys, which is somewhat confusing, as the term typically refers to the D-isomer of cysteine. Please clarify whether the amine group was specifically removed or if the entire amino acid was replaced or modified.

We used 3-mercaptopropionic acid to introduce the N-terminal deaminocysteine (dCys). We have now defined this better and included it in the Experimental section.

"...and Fmoc-protected S-Trityl-3-mercaptopropionic acid ((Fmoc-dCys(Trt))-OH) were purchased..."

2. The authors have used HPLC to determine the hydrophobicity of oxytocin derivatives. However, hydrophobicity is typically assessed using the water-octanol partitioning system. It would be more informative to include cLogP determination using the water-octanol system for a more conventional and standardized evaluation of hydrophobicity.

We thank the reviewer for this suggestion. We have now included calculated LogP (cLogP) values for all analogues in SI (shown below). While cLogP is a highly useful computational estimate of hydrophobicity based on molecular structure for small molecules, it often does not reflect well the actual hydrophobicity of peptides, primarily due to the specific secondary structure, hydrogen bonding, disulfide macrocyclization, charge distribution, and buried residues that are not taken into account in cLogP determination. This can also be observed in our study, where not all of the experimentally determined retention times align with the cLogP values order. For these reasons, we still prioritise the experimentally determined HPLC retention time as our primary readout; however, we have now also included the cLogP values as we agree that this may be of interest to the reader. We added the following:

Analogue	Retention time	cLogP
OTK⁸[Aoa-FDG]	23 min	-4.235
dOTK⁸[ESF]	24 min	-1.456
OT	27 min	-0.654
dOTK⁸[SFB]	29 min	0.134
dOTK⁸[ESF₂]	33 min	-0.671

“...this order is slightly different from the one obtained using cLogP values, as it takes into account secondary structure, hydrogen bonding, disulfide macrocyclization, charge distribution and buried residues.”

“Comparison of calculated LogP (cLogP) values and experimentally determined HPLC retention times for the OT analogues. While cLogP provides a theoretical estimate of hydrophobicity based on molecular structure, HPLC retention time offers an experimentally derived measure that accounts for the full physicochemical behaviour of the peptides under chromatographic conditions.”

3. The manuscript begins by discussing intranasal administration but ultimately focuses on intravenous (IV) administration. Have the authors attempted intranasal administration of the radiotracer? If so, how does its pharmacokinetics compare to intravenous administration? Providing data or discussion on this aspect would enhance the relevance of the study to intranasal delivery applications.

This work is part of a broader effort preparing for human studies that will include both intravenous (IV) and intranasal (IN) administration (also stated in the conclusions). We have not performed IN administration studies in our rodent model due to significant anatomical and physiological differences in nasal structure and function between rodents and humans. These differences, along with variations in administration techniques and uncertainties in dosimetry for small animals, make direct translational conclusions from rodent IN studies particularly challenging. We therefore prioritised in this study IV to establish a more robust and reproducible baseline for tracer administration and distribution, demonstrating that our tracer can be imaged and collecting preliminary imaging data on its interaction with OTR-rich tissue. IN and IV administration of our lead tracer is planned for human studies to assess OT's biodistribution and potential brain uptake.

Reviewer #3 (Remarks to the Author):

Review and comments to the authors

The authors present the synthesis and pharmacological characterization of five oxytocin-like peptides and selected the most suitable candidate for development as fluorine-18 labeled radiotracer for preclinical investigation of the oxytocin/oxytocin receptor signaling system by PET imaging.

In my opinion the manuscript is very well written, the content is within the scope of the journal and will be of interest to scientist working in the field of PET radiotracer development. The experiments are well designed and the presented results are supported by the data obtained. Therefore, I propose to publish the article after some minor corrections.

We thank the reviewer for taking the time to go through the manuscript and for the useful feedback and comments, which we have addressed in the revised manuscript.

General remark:

There are only a few references in the manuscript directing the reader to the Supplementary Information and most of them are incorrect (e.g. line 204, line 217, line 428). As the content of the SI is important for the manuscript, I would like to recommend to refer to each section and each figure presented in the SI at an appropriate position in the manuscript.

We thank the reviewer for pointing out this issue.

We have reviewed all the cross-references and amended the wrong ones.

Detailed remarks:

- line 117: The five peptides shown in Figure 1A are not yet tracers, same in line 167.

We have modified the term “tracer” on this and other occasions.

- line 162: Please add vasopressin for VP.

Vasopressin (VP) was added.

- line 165-166: Please briefly explain the principle of the assay.

The FLIPR (Fluorescence Imaging Plate Reader) assay measures intracellular calcium flux as a readout of GPCR activation. Upon ligand binding, G protein-coupled receptors (such as OTR) trigger calcium release from intracellular stores, which is detected using a fluorescent calcium-sensitive dye. The resulting fluorescence signal provides a quantitative measure of receptor activation in real time.

- line 167: I think you cannot state here, that a potency of 5-47 nM is sufficient for PET imaging, because that is what was actually also investigated in this study.

What we meant was that the observed potency range lies within values typically considered compatible with PET tracer development, based on prior literature. We rephrased this sentence for better clarity:

“All tested tracers retained low nanomolar potency at OTR (5–47 nM), a range consistent with values reported for other validated PET tracers.”

- line 193: The authors mention the "direct labeling of OT". Please include and name this radiotracer in Figure 1 or as a new scheme in the SI.

We added a new scheme in the SI.

“This section reports example HPLC profiles of the reaction mixtures for the radioconjugation reactions shown below.”

-Figure 3 A and B: It is not the RCY of OT or dOTK8 shown but of the corresponding ^{18}F -labeled tracers. Please correct the axis labels.

We thank the reviewer for pointing this out. We intended to indicate the precursors rather than the product in the labels. We have now clarified this in the caption title and in the graph y-axis description.

“Fig. 1: Optimization of radiolabelling of OT and dOTK⁸ precursors with ^{18}F SFB.”

- lines 251-253: I suggest adding a sentence describing the observation of differences in the elution order of F2 and F3 in the semi-prep and analytical chromatograms.

We added a statement reporting this observation.

"...also indicating the relation of the collected fractions with the analytical chromatogram of the mixture (Fig. 4B), and noticing an inversion of R_t for fractions F2 and F3 between the two chromatographic systems."

- Figure 4: Please include the conditions for semi-prep HPLC and analytical HPLC in the Figure caption.

We included the semi-prep conditions in the caption of Figure 4; the analytical conditions are indicated in the caption and explained in the Experimental section.

"...(Chromolith RP semi-prep column (150 x 10mm, 13 nm pores), using a 25/75=ACN/H₂O with 0.05% TFA (v/v) at a 2.5mL/min flow rate),..."

- lines 267-269: I recommend to include here information on the sorption and recovery to demonstrate that there is no considerable loss of activity due to the simplified process.

We thank the reviewer for this suggestion. We have now included quantitative data on product recovery and sorption losses in the revised manuscript. The simplified formulation process resulted in a recovery of $62 \pm 7\%$ of the collected radioactive fraction. Losses due to cartridge breakthrough (i.e., insufficient retention) were measured at $14 \pm 8\%$, which is consistent with values reported for similar tracers and formulation setups.

"This simplified formulation process resulted in a recovery of $62 \pm 7\%$ of the collected radioactive product, with $14 \pm 8\%$ lost due to insufficient retention during cartridge loading (i.e., breakthrough)."

- line 302: According to the nomenclature rules published in 2017 (<https://doi.org/10.1016/j.nucmedbio.2017.09.004>), I recommend to use A_m instead of A_M as abbreviation for molar activity. Please change it accordingly throughout the manuscript and SI.

We thank the reviewer for this note; we have adopted this nomenclature in the revised manuscript.

- SI, Section 3, "Bioconjugation experiments using [¹⁸F]ESF". I think you mean [¹⁸F]SFB? Please change in the whole chapter.

In this section, we indeed used [¹⁸F]ESF, as anticipated in the text at lines 188-189 of the original manuscript.

No changes were necessary.

- SI, Section 6, "Calibration curve for [18F]dOTK8[SFB]". This is not a calibration curve for the radiotracer but for the non-radioactive compound. Please correct accordingly.

We removed [¹⁸F].

- SI, Chromatograms SuppInfo 8-11: Please indicate the desired products in the chromatograms.

We revised the captions of all the chromatograms to indicate the identified radioproducts.